# `T2V-Turbo`: Breaking the Quality Bottleneck of Video Consistency Model with Mixed Reward Feedback

**Jiachen Li**[1], **Weixi Feng**[1], **Tsu-Jui Fu**[1], **Xinyi Wang**[1], **Sugato Basu**[2],
**Wenhu Chen**[3], **William Yang Wang**[1]
[1]UC Santa Barbara, [2]Google, [3]University of Waterloo
[1]{jiachen_li, weixifeng, tsu-juifu, xinyi_wang, william}@cs.ucsb.edu
[2]sugato@google.com   [3]wenhuchen@uwaterloo.ca

Project Page: `https://t2v-turbo.github.io`

**Videos: click to play in Adobe Acrobat**

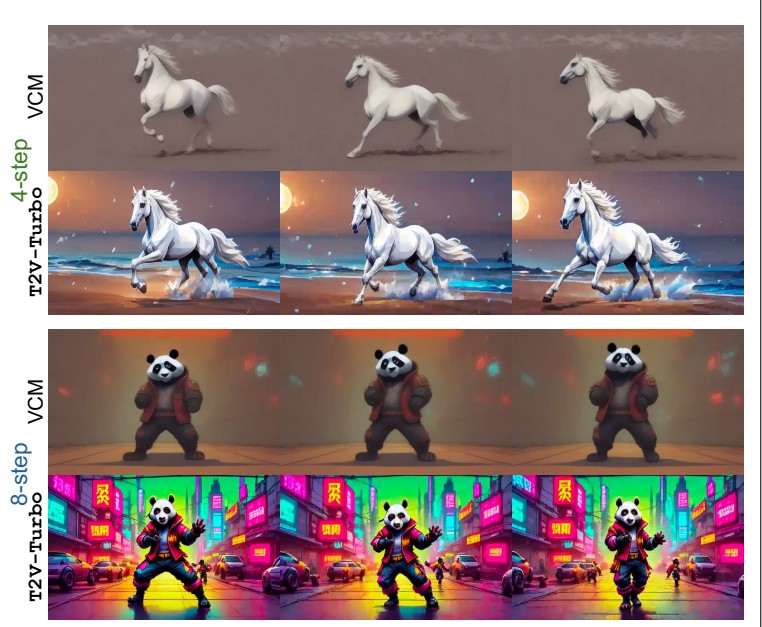

Figure 1: By integrating reward feedback during consistency distillation from VideoCrafter2 [Chen et al., 2024], our `T2V-Turbo` (VC2) can generate high-quality videos with 4-8 inference steps, breaking the quality bottleneck of a VCM [Wang et al., 2023a]. Appendix F includes the corresponding text prompts.

## Abstract

Diffusion-based text-to-video (T2V) models have achieved significant success but continue to be hampered by the slow sampling speed of their iterative sampling processes. To address the challenge, consistency models have been proposed to facilitate fast inference, albeit at the cost of sample quality. In this work, we aim to break the quality bottleneck of a video consistency model (VCM) to achieve **both fast and high-quality video generation**. We introduce `T2V-Turbo`, which integrates feedback from a mixture of differentiable reward models into the consistency distillation (CD) process of a pre-trained T2V model. Notably, we directly optimize rewards associated with single-step generations that arise naturally from computing the CD loss, effectively bypassing the memory constraints imposed by backpropagating gradients through an iterative sampling process. Remarkably, the 4-step generations from our `T2V-Turbo` achieve the highest total score on VBench [Huang et al., 2024], even surpassing Gen-2 [Esser et al., 2023] and

Pika [Pika Labs, 2023]. We further conduct human evaluations to corroborate the results, validating that the 4-step generations from our T2V-Turbo are preferred over the 50-step DDIM samples from their teacher models, representing more than a tenfold acceleration while improving video generation quality.

# 1 Introduction

Diffusion model (DM) [Sohl-Dickstein et al., 2015, Ho et al., 2020] has emerged as a powerful framework for neural image [Betker et al., 2023, Rombach et al., 2022, Esser et al., 2024, Saharia et al., 2022] and video synthesis [Singer et al., 2022, Ho et al., 2022a, He et al., 2022, Wang et al., 2023b, Zhang et al., 2023], leading to the development of cutting-edge text-to-video (T2V) models like Sora [Brooks et al., 2024], Gen-2 [Esser et al., 2023] and Pika [Pika Labs, 2023]. Although the iterative sampling process of these diffusion-based models ensures high-quality generation, it significantly slows down inference, hindering their real-time applications. On the other hand, existing open-sourced T2V models including VideoCrafter [Chen et al., 2023, 2024] and ModelScopeT2V [Wang et al., 2023c] are trained on web-scale video datasets, e.g., WebVid-10M [Bain et al., 2021], with varying video qualities. Consequently, the generated videos often appear visually unappealing and fail to align accurately with the text prompts, deviating from human preferences.

Efforts have been made to address the issues listed above. To accelerate the inference process, Wang et al. [2023a] applies the theory of consistency distillation (CD) [Song et al., 2023, Song and Dhariwal, 2023, Luo et al., 2023a] to distill a video consistency model (VCM) from a teacher T2V model, enabling plausible video generations in just 4-8 inference steps. However, the quality of VCM's generations is naturally bottlenecked by the performance of the teacher model, and the reduced number of inference steps further diminishes its generation quality. On the other hand, to align generated videos with human preferences, InstructVideo [Yuan et al., 2023] draws inspiration from image generation techniques [Dong et al., 2023, Clark et al., 2023, Prabhudesai et al., 2023] and proposes backpropagating the gradients of a differentiable reward model (RM) through the iterative video sampling process. However, calculating the full reward gradient is prohibitively expensive, resulting in substantial memory costs. Consequently, InstructVideo truncates the sampling chain by limiting gradient calculation to only the final DDIM step, compromising optimization accuracy. Additionally, InstructVideo is limited by its reliance on an image-text RM, which fails to fully capture the transition dynamic of a video. Empirically, InstructVideo only conducts experiments on a limited set of user prompts, the majority of which are related to animals. As a result, its generalizability to a broader range of prompts remains unknown.

In this paper, we aim to achieve fast and high-quality video generation by breaking the quality bottleneck of a VCM. We introduce T2V-Turbo, which integrates reward feedback from a mixture of RMs into the process of distilling a VCM from a teacher T2V model. Besides utilizing an image-text RM to align individual video frames with human preference, we further incorporate reward feedback

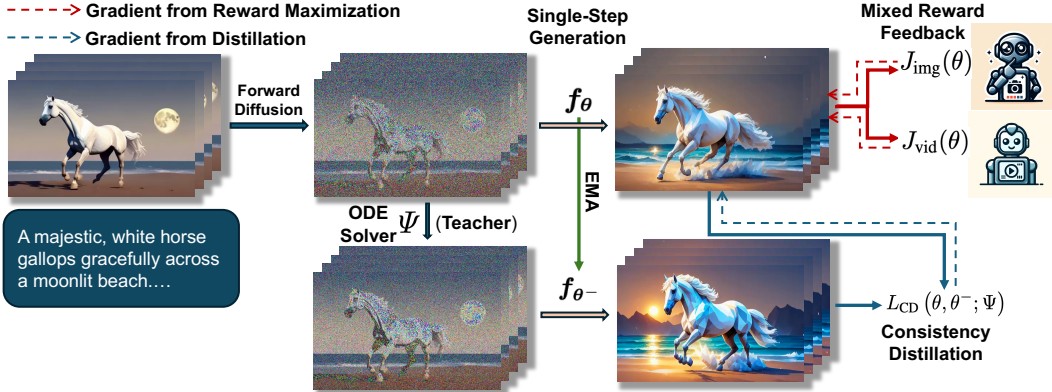

Figure 2: Overview of the training pipeline of our T2V-Turbo. We integrate reward feedback from both an image-text RM and a video-text RM into the VCD procedures by backpropagating gradient through the single-step generation process of our T2V-Turbo.

from a video-text RM to comprehensively evaluate the temporal dynamics and transitions in the generated videos. We highlight that our reward optimization avoids tackling the highly memory-intensive issues associated with backpropagating gradients through an iterative sampling process. Instead, we directly optimize rewards of the single-step generations that arise from computing the CD loss, effectively bypassing the memory constraints faced by conventional methods that optimize a DM [Yuan et al., 2023, Xu et al., 2024a, Clark et al., 2023, Prabhudesai et al., 2023].

Empirically, we demonstrate the superiority of our `T2V-Turbo` in generating high-quality videos within 4-8 inference steps. To illustrate the applicability of our methods, we distill `T2V-Turbo` (VC2) and `T2V-Turbo` (MS) from VideoCrafter2 [Chen et al., 2024] and ModelScopeT2V [Wang et al., 2023c], respectively. Remarkably, the 4-step generation results from both variants of our `T2V-Turbo` outperform SOTA models on the video evaluation benchmark VBench [Huang et al., 2024], even surpassing proprietary systems such as Gen-2 [Esser et al., 2023] and Pika [Pika Labs, 2023] that are trained with extensive resources. We further corroborate the results by conducting human evaluation using 700 prompts from the EvalCrafter [Liu et al., 2023] benchmark, validating that the 4-step generations from `T2V-Turbo` are favored by human over the 50-step DDIM samples from their teacher T2V models, which represents over tenfold inference acceleration and enhanced video generation quality.

Our contributions are threefold:

- Learn a T2V model with feedback from a mixture of RMs, including a video-text model. To the best of our knowledge, we are the first to do so.
- Establish a new SOTA on the VBench with only 4 inference steps, outperforming proprietary models trained with substantial resources.
- 4-step generations from our `T2V-Turbo` are favored over the 50-step generation from its teacher T2V model as evidenced by human evaluation, representing over 10 times inference acceleration with quality improvement.

## 2 Preliminaries

**Diffusion models (DMs)**. In the forward process, DMs progressively inject Gaussian noise into the original data distribution $p_{\text{data}}(\mathbf{x}) \equiv p_0(\mathbf{x}_0)$ and perturb it into a marginal distribution $p_t(\mathbf{x}_t)$ with the transition kernel $p_{0t}(\mathbf{x}_t|\mathbf{x}_0) = \mathcal{N}(\mathbf{x}_t|\alpha(t)\mathbf{x}_0, \beta^2(t)\mathbf{I})$ at timestep $t$. $\alpha(t)$ and $\beta(t)$ correspond to the noise schedule. In the reverse process, DMs sequentially recover the data from a noise sampled from the prior distribution $p_T(\mathbf{x}_T) := \mathcal{N}(\mathbf{x}_T|\mathbf{0}, \beta^2(T)\mathbf{I})$. The reverse-time SDE can be modeled by an ordinary differential equation (ODE), known as the Probability Flow (PF-ODE) [Song et al., 2020a]:

$$d\mathbf{x}_t = \left[\boldsymbol{\mu}\left(t\right)\mathbf{x}_t - \frac{1}{2}\sigma(t)^2\nabla\log p_t\left(\mathbf{x}_t\right)\right]dt, \quad \mathbf{x}_T \sim \mathcal{N}(\mathbf{0}, \beta^2(T)\mathbf{I}). \tag{1}$$

where $\boldsymbol{\mu}(\cdot)$ and $\sigma(\cdot)$ are the drift and diffusion coefficients, respectively, with the following properties:

$$\boldsymbol{\mu}(t) = \frac{d\log\alpha(t)}{dt}, \quad \sigma^2(t) = \frac{d\beta^2(t)}{dt} - 2\frac{d\log\alpha(t)}{dt}\beta^2(t). \tag{2}$$

The PF-ODE's solution trajectories, when sampled at any timestep $t$, align with the distribution $p_t(\mathbf{x}_t)$. Empirically, a denoising model $\epsilon_\theta(\mathbf{x}_t, t)$ is trained to approximate the score function $-\nabla\log p_t(\mathbf{x}_t)$ via score matching. During the sampling phase, one begins with a sample $\mathbf{x}_T \sim p_T(\mathbf{x}_T)$ and follows the empirical PF-ODE below to obtain a sample $\hat{\mathbf{x}}_0$.

$$d\mathbf{x}_t = \left[\boldsymbol{\mu}\left(t\right)\mathbf{x}_t + \frac{1}{2}\sigma(t)^2\boldsymbol{\epsilon}_\theta(\mathbf{x}_t, t)\right]dt, \quad \mathbf{x}_T \sim \mathcal{N}(\mathbf{0}, \beta^2(T)\mathbf{I}). \tag{3}$$

In this paper, we focus on diffusion-based T2V models, which operate on the video latent space $\mathcal{Z}$ and train a denoising model $\epsilon_\theta(\mathbf{z}_t, \mathbf{c}, t)$ conditioned on the text prompt $\mathbf{c}$, where $\mathbf{z}_t$ is obtained by perturbing the image latent $\mathbf{z} = \mathcal{E}(\mathbf{x}), \in \mathcal{Z}$ and $\mathcal{E}$ is a VAE [Kingma and Welling, 2013] encoder. The T2V models employ Classifier-Free Guidance (CFG) [Ho and Salimans, 2021] to enhance the quality of conditional sampling by substituting the noise prediction with a linear combination of conditional and unconditional noise predictions for denoising, i.e., $\tilde{\epsilon}_\theta(\mathbf{z}_t, \omega, \mathbf{c}, t) = (1+\omega)\epsilon_\theta(\mathbf{z}_t, \mathbf{c}, t) - \omega\epsilon_\theta(\mathbf{z}_t, \varnothing, t)$, where $\omega$ is the CFG scale. After the completion of the inference process, we can generate a video by $\hat{\mathbf{x}}_0 = \mathcal{D}(\mathbf{z}_0)$ with the VAE decoder $\mathcal{D}$ corresponding to $\mathcal{E}$.

**Consistency Distillation.** Conventional methods [Ho et al., 2020, Song et al., 2020b] generate their samples by solving the PF-ODE sequentially, leading to DM's slow inference speed. To tackle this problem, *consistency models* (CM) [Song et al., 2023, Song and Dhariwal, 2023] propose to learn a consistency function $\boldsymbol{f} : (\mathbf{x}_t, t) \mapsto \mathbf{x}_\epsilon$ to directly map any $\mathbf{x}_t$ on the PF-ODE trajectory to its origin, where $\epsilon$ is a fixed small positive number. And thus, the consistency function $\boldsymbol{f}$ has the following *self-consistency* property

$$\boldsymbol{f}(\mathbf{x}_t, t) = \boldsymbol{f}(\mathbf{x}'_t, t'), \forall t, t' \in [\epsilon, T], \tag{4}$$

where $\mathbf{x}_t$ and $\mathbf{x}'_t$ are from the same PF-ODE. We can model $\boldsymbol{f}$ with a CM $\boldsymbol{f}_\theta$. When tackling the PF-ODE of a T2V model that operates on the video latent space $\mathcal{Z}$, we aim to learn a video consistency model (VCM) [Luo et al., 2023a, Wang et al., 2023a] $\boldsymbol{f}_\theta : (\mathbf{z_t}, \omega, \boldsymbol{c}, t) \mapsto \mathbf{z_0} \in \mathcal{Z}$. To ensure $\boldsymbol{f}_\theta(\mathbf{z}, \omega, \boldsymbol{c}, t) = \mathbf{z}$, we parameterize $\boldsymbol{f}_\theta$ as

$$\boldsymbol{f}_\theta(\mathbf{z}, \omega, \boldsymbol{c}, t) = c_{\text{skip}}(t)\mathbf{z} + c_{\text{out}}(t)F_\theta(\mathbf{z}, \omega, \boldsymbol{c}, t), \tag{5}$$

where $c_{\text{skip}}(t)$ and $c_{\text{out}}(t)$ are differentiable functions with $c_{\text{skip}}(\epsilon) = 1$ and $c_{\text{out}}(\epsilon) = 0$, and $F_\theta$ is modeled as a neural network. We can distill a $\boldsymbol{f}_\theta$ from a pre-trained T2V DM by minimizing the *consistency distillation* (CD) [Song et al., 2023, Luo et al., 2023a] loss as below

$$L_{\text{CD}}\left(\theta, \theta^-; \Psi\right) = \mathbb{E}_{\mathbf{z}, \mathbf{c}, \omega, n}\left[d\left(\boldsymbol{f}_\theta\left(\mathbf{z}_{t_{n+k}}, \omega, \mathbf{c}, t_{n+k}\right), \boldsymbol{f}_{\theta^-}\left(\hat{\mathbf{z}}_{t_n}^{\Psi, \omega}, \omega, \mathbf{c}, t_n\right)\right)\right], \tag{6}$$

where $d(\cdot, \cdot)$ is a distance function. $\theta^-$ is updated by the exponential moving average (EMA) of $\theta$, i.e., $\theta^- \leftarrow \texttt{stop\_grad}\left(\mu\theta + (1 - \mu)\theta^-\right)$. $\hat{\mathbf{z}}_{t_n}^{\Psi, \omega}$ is an estimate of $\mathbf{z}_{t_n}$ obtained by the numerical augmented PF-ODE solver $\Psi$ parameterized by $\psi$ and $k$ is the skipping interval

$$\hat{\mathbf{z}}_{t_n}^{\Psi, \omega} \leftarrow \mathbf{z}_{t_{n+k}} + (1 + \omega)\Psi(\mathbf{z}_{t_{n+k}}, t_{n+k}, t_n, \boldsymbol{c}; \psi) - \omega\Psi(\mathbf{z}_{t_{n+k}}, t_{n+k}, t_n, \varnothing; \psi). \tag{7}$$

We follow the LCM paper [Luo et al., 2023a] to use DDIM [Song et al., 2020b] as the ODE solver $\Psi$ and defer the formula of the DDIM solver to Appendix A.

## 3   Training `T2V-Turbo` with Mixed Reward Feedback

In this section, we present the training pipeline to derive our `T2V-Turbo`. To facilitate fast and high-quality video generation, we integrate reward feedback from multiple RMs into the LCD process when distilling from a teacher T2V model. Figure 2 provides an overview of our framework. Notably, we directly leverage the single-step generation $\hat{\mathbf{z}}_0 = \boldsymbol{f}_\theta\left(\mathbf{z}_{t_{n+k}}, \omega, \mathbf{c}, t_{n+k}\right)$ arise from computing the CD loss $L_{\text{CD}}$ (6) and optimize the video $\hat{\mathbf{x}}_0 = \mathcal{D}(\hat{\mathbf{z}}_0)$ decoded from it towards multiple differentiable RMs. As a result, we avoid the challenges associated with backpropagating gradients through an iterative sampling process, which is often confronted by conventional methods optimizing DMs [Clark et al., 2023, Xu et al., 2024a, Yuan et al., 2023].

In particular, we leverage reward feedback from an image-text RM to improve human preference on each individual video frame (Sec. 3.1) and further utilize the feedback from a video-text RM to improve the temporal dynamics and transitions in the generated video (Sec. 3.2).

### 3.1   Optimizing Human Preference on Individual Video Frames

Chen et al. [2024] achieve high-quality video generation by including high-quality images as single-frame videos when training the T2V model. Inspired by their success, we align each individual video frame with human preference by optimizing towards a differentiable image-text RM $\mathcal{R}_{\text{img}}$. In particular, we randomly sample a batch of $M$ frames $\{\hat{\mathbf{x}}_0^1, \dots, \hat{\mathbf{x}}_0^M\}$ from the decoded video $\hat{\mathbf{x}}_0$ and maximize their scores evaluated by $\mathcal{R}_{\text{img}}$ as below

$$J_{\text{img}}(\theta) = \mathbb{E}_{\hat{\mathbf{x}}_0, \mathbf{c}}\left[\sum_{m=1}^M \mathcal{R}_{\text{img}}\left(\hat{\mathbf{x}}_0^m, \mathbf{c}\right)\right], \quad \hat{\mathbf{x}}_0 = \mathcal{D}\left(\boldsymbol{f}_\theta\left(\mathbf{z}_{t_{n+k}}, \omega, \mathbf{c}, t_{n+k}\right)\right). \tag{8}$$

### 3.2   Optimizing Video-Text Feedback Model

Existing image-text RMs [Wu et al., 2023a, Xu et al., 2024a, Kirstain et al., 2024] are limited to assessing the alignment between individual video frames and the text prompt and thus cannot evaluate

through the temporal dimensions that involve inter-frame dependencies, such as motion dynamic and transitions [Huang et al., 2024, Liu et al., 2023]. To address these shortcomings, we further leverage a video-text RM $\mathcal{R}_{\text{vid}}$ to assess the generated videos. The corresponding objective $J_{\text{vid}}$ is given below

$$J_{\text{vid}}(\theta) = \mathbb{E}_{\hat{\mathbf{x}}_0, \mathbf{c}} \left[ \mathcal{R}_{\text{vid}} \left( \hat{\mathbf{x}}_0, \mathbf{c} \right) \right], \quad \hat{\mathbf{x}}_0 = \mathcal{D} \left( \boldsymbol{f}_\theta \left( \mathbf{z}_{t_{n+k}}, \omega, \mathbf{c}, t_{n+k} \right) \right). \tag{9}$$

### 3.3 Summary

To this end, we can define the total learning loss $L$ of our training pipeline as a linear combination of the $L_{\text{CD}}$ in (6), $J_{\text{img}}$ in (8), and $J_{\text{vid}}$ in (9) with weighting parameters $\beta_{\text{img}}$ and $\beta_{\text{vid}}$.

$$L \left( \theta, \theta^-; \Psi \right) = L_{\text{CD}} \left( \theta, \theta^-; \Psi \right) - \beta_{\text{img}} J_{\text{img}}(\theta) - \beta_{\text{vid}} J_{\text{vid}}(\theta) \tag{10}$$

To reduce memory and computational cost, we initialize our T2V-Turbo with the teacher model and only optimize the LoRA weights [Hu et al., 2021, Luo et al., 2023b] instead of performing full model training. After completing the training, we merge the LoRA weights so that the per-step inference cost of our T2V-Turbo remains identical to the teacher model. We include pseudo-codes for our training algorithm in Appendix B.

## 4 Experimental Results

Our experiments aim to demonstrate our T2V-Turbo's ability to generate high-quality videos with 4-8 inference steps. We first conduct automatic evaluations on the standard benchmark VBench [Huang et al., 2024] to comprehensively evaluate our methods from various dimensions (Sec. 4.1) against a broad array of baseline methods. We then perform human evaluations with 700 prompts from the EvalCrafter [Liu et al., 2023] to compare the 4-step and 8-step generations from our T2V-Turbo with the 50-step generations from the teacher T2V models as well as the 4-step generations from the baseline VCM (Sec. 4.2). Finally, we perform ablation studies on critical design choices (Sec. 4.3).

**Settings**. We train T2V-Turbo (VC2) and T2V-Turbo (MS) by distilling from the teacher diffusion-based T2V models VideoCrafter2 [Chen et al., 2024] and ModelScopeT2V [Wang et al., 2023c], respectively. Similar to both teacher models, we conduct our training using the WebVid10M [Bain et al., 2021] datasets. We train our models on 8 NVIDIA A100 GPUs for 10K gradient steps without gradient accumulation. We set the batch size of training videos to 1 for each GPU device. We employ HPSv2.1 [Wu et al., 2023a] as our image-text RM $\mathcal{R}_{\text{img}}$. When distilling from VideoCrafter2, we utilize the 2nd Stage model of InternVideo2 (InternVid2 S2) [Wang et al., 2024] as our video-text RM $\mathcal{R}_{\text{vid}}$. When distilling from ModelScopeT2V, we set $\mathcal{R}_{\text{vid}}$ to be ViCLIP [Wang et al., 2023d]. To optimize $J_{\text{img}}$ (8), we randomly sample 6 frames from the video by setting $M = 6$. For the hyperparameters (HP), we set learning rate $1e-5$ and guidance scale range $[\omega_{\min}, \omega_{\max}] = [5, 15]$. We use DDIM [Song et al., 2020b] as our ODE solver $\Psi$ and set the skipping step $k = 20$. For T2V-Turbo (VC2), we set $\beta_{\text{img}} = 1$ and $\beta_{\text{vid}} = 2$. For T2V-Turbo (MS), we set $\beta_{\text{img}} = 2$ and $\beta_{\text{vid}} = 3$. We include further training details in Appendix A.

### 4.1 Automatic Evaluation on VBench

We evaluate our T2V-Turbo (VC2) and T2V-Turbo (MS) on the standard video evaluation benchmark VBench [Huang et al., 2024] to compare against a wide array of baseline methods. VBench is designed to comprehensively evaluate T2V models from 16 disentangled dimensions. Each dimension in VBench is tailored with specific prompts and evaluation methods.

Table 1 compares the 4-step generation of our methods with various baselines from the VBench leaderboard[1], including Gen-2 [Esser et al., 2023], Pika [Pika Labs, 2023], VideoCrafter1 [Chen et al., 2023], VideoCrafter2 [Chen et al., 2024], Show-1 [Zhang et al., 2023], LaVie [Wang et al., 2023b], and ModelScopeT2V [Wang et al., 2023c]. Table 4 in Appendix further compares our methods with VideoCrafter0.9 [He et al., 2022], LaVie-Interpolation [Wang et al., 2023b], Open-Sora [Open-Sora, 2024], and CogVideo [Hong et al., 2022]. The performance of each baseline method is directly reported from the VBench leaderboard. To obtain the results of our methods, we carefully follow VBench's evaluation protocols by generating 5 videos for each prompt to calculate the metrics. We

---

[1]`https://huggingface.co/spaces/Vchitect/VBench_Leaderboard`

Table 1: **Automatic Evaluation on VBench** [Huang et al., 2024]. We compare our `T2V-Turbo` (VC2) and `T2V-Turbo` (MS) with baseline methods across the 16 VBench dimensions. A higher score indicates better performance for a particular dimension. We bold the best results for each dimension and underline the second-best result. **Quality Score** is calculated with the 7 dimensions from the top table. **Semantic Score** is calculated with the 9 dimensions from the bottom table. **Total Score** a weighted sum of **Quality Score** and **Semantic Score**. Further details can be found in Appendix C. Both our `T2V-Turbo` (VC2) and `T2V-Turbo` (MS) **surpass all baseline methods with 4 inference steps** in terms of Total Score, including the proprietary systems Gen-2 and Pika.

| Models | Total Score | Quality Score | Subject Consist. | BG Consist. | Temporal Flicker. | Motion Smooth. | Aesthetic Quality | Dynamic Degree | Image Quality |
|---|---|---|---|---|---|---|---|---|---|
| ModelScopeT2V | 75.75 | 78.05 | 89.87 | 95.29 | 98.28 | 95.79 | 52.06 | **66.39** | 58.57 |
| LaVie | 77.08 | 78.78 | 91.41 | 97.47 | 98.30 | 96.38 | 54.94 | 49.72 | 61.90 |
| Show-1 | 78.93 | 80.42 | 95.53 | 98.02 | 99.12 | 98.24 | 57.35 | 44.44 | 58.66 |
| VideoCrafter1 | 79.72 | 81.59 | 95.10 | 98.04 | 98.93 | 95.67 | 62.67 | 55.00 | 65.46 |
| Pika | 80.40 | **82.68** | 96.76 | **98.95** | **99.77** | 99.51 | 63.15 | 37.22 | 62.33 |
| VideoCrafter2 | 80.44 | 82.20 | 96.85 | 98.22 | 98.41 | 97.73 | 63.13 | 42.50 | 67.22 |
| Gen-2 | 80.58 | 82.47 | **97.61** | 97.61 | 99.56 | **99.58** | **66.96** | 18.89 | 67.42 |
| VCM (MS) | 75.84 | 78.80 | 93.06 | 97.30 | 98.51 | 98.00 | 48.99 | 46.11 | 61.98 |
| Our `T2V-Turbo` (MS) | 80.62 | 82.15 | 94.82 | 98.71 | 97.99 | 95.64 | 60.04 | **66.39** | 68.09 |
| VCM (VC2) | 73.97 | 78.54 | 94.02 | 96.05 | 99.06 | 98.84 | 54.56 | 42.50 | 52.72 |
| Our `T2V-Turbo` (VC2) | **81.01** | 82.57 | 96.28 | 97.02 | 97.48 | 97.34 | 63.04 | 49.17 | **72.49** |

| Models | Semantic Score | Object Class | Multiple Objects | Human Action | Color | Spatial Relation. | Scene | Appear. Style | Temporal Style | Overall Consist. |
|---|---|---|---|---|---|---|---|---|---|---|
| ModelScopeT2V | 66.54 | 82.25 | 38.98 | 92.40 | 81.72 | 33.68 | 39.26 | 23.39 | 25.37 | 25.67 |
| LaVie | 70.31 | 91.82 | 33.32 | **96.80** | 86.39 | 34.09 | 52.69 | 23.56 | 25.93 | 26.41 |
| Show-1 | 72.98 | 93.07 | 45.47 | 95.60 | 86.35 | 53.50 | 47.03 | 23.06 | 25.28 | 27.46 |
| VideoCrafter1 | 72.22 | 78.18 | 45.66 | 91.60 | **93.32** | 58.86 | 43.75 | 24.41 | 25.54 | 26.76 |
| Pika | 71.26 | 87.45 | 46.69 | 88.00 | 85.31 | 65.65 | 44.80 | 21.89 | 24.44 | 25.47 |
| VideoCrafter2 | 73.42 | 92.55 | 40.66 | 95.00 | 92.92 | 35.86 | 55.29 | 25.13 | 25.84 | **28.23** |
| Gen-2 | 73.03 | 90.92 | 55.47 | 89.20 | 89.49 | **66.91** | 48.91 | 19.34 | 24.12 | 26.17 |
| VCM (MS) | 63.98 | 83.18 | 24.85 | 87.20 | 85.72 | 31.57 | 42.44 | 23.20 | 23.30 | 24.18 |
| Our `T2V-Turbo` (MS) | 74.47 | 93.34 | **58.63** | 95.80 | 89.67 | 45.74 | 48.47 | 23.23 | 25.92 | 27.51 |
| VCM (VC2) | 55.66 | 63.97 | 10.81 | 82.60 | 79.12 | 23.06 | 18.49 | **25.29** | 22.31 | 25.15 |
| Our `T2V-Turbo` (VC2) | **74.76** | **93.96** | 54.65 | 95.20 | 89.90 | 38.67 | **55.58** | 24.42 | 25.51 | 28.16 |

further train VCM (VC2) and VCM (MS) by distilling from VideoCrafter2 and ModelScopeT2V, respectively, without incorporating reward feedback, and then compare their results.

VBench has developed its own rules to calculate the **Total Score**, **Quality Score**, and **Semantic Score**. **Quality Score** is calculated with the 7 dimensions from the top table. **Semantic Score** is calculated with the 9 dimensions from the bottom table. And **Total Score** is a weighted sum of Quality Score and Semantic Score. Appendix C provides further details, including explanations for each dimension of VBench. As shown in Table 1, the 4-step generations of both our `T2V-Turbo` (MS) and `T2V-Turbo` (VC2) surpass all baseline methods on VBench in terms of Total Score. These results are particularly remarkable given that we even outperform the proprietary systems Gen-2 and Pika, which are trained with extensive resources. Even when distilling from a less advanced teacher model, ModelScopeT2V, our `T2V-Turbo` (MS) attains the second-highest Total Score, just below our `T2V-Turbo` (VC2). Additionally, our `T2V-Turbo` breaks the quality bottleneck of a VCM by outperforming its teacher T2V model, significantly improving over the baseline VCM.

### 4.2 Human Evaluation with 700 EvalCrafter Prompts

To verify the effectiveness of our `T2V-Turbo`, we compare the 4-step and 8-step generations from our `T2V-Turbo` with the 50-step DDIM samples from the corresponding teacher T2V models. We further compare the 4-step generations between our `T2V-Turbo` and their baseline VCMs when distilled from the same teacher T2V model. We leverage the 700 prompts from the EvalCrafter [Liu et al., 2023] video evaluation benchmark, which are constructed based on real-world user data.

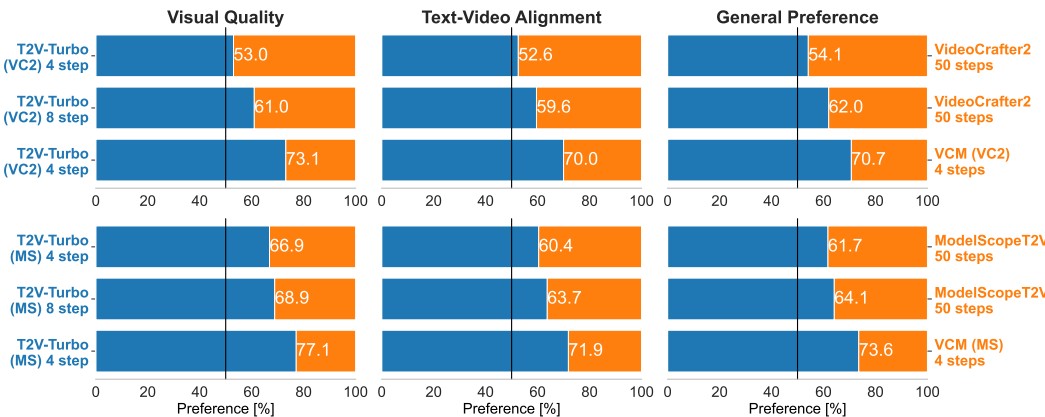

Figure 3: Human evaluation results with the 700 prompts from EvalCrafter [Liu et al., 2023]. We compare the 4-step and 8-step generations from our T2V-Turbo with their teacher T2V model and their baseline VCM. **Top**: results for T2V-Turbo (VC2). **Bottom**: results for T2V-Turbo (MS).

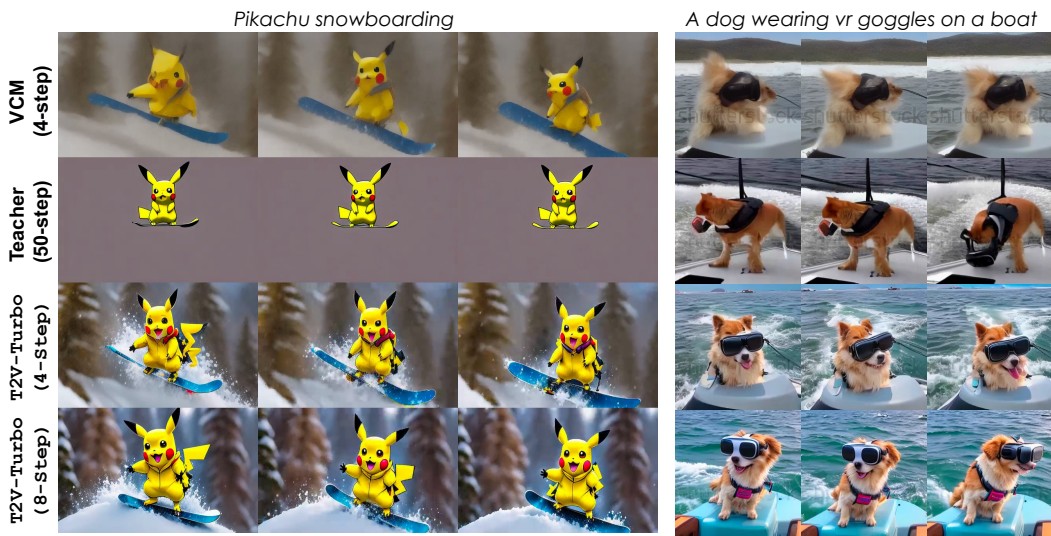

Figure 4: Qualitative comparisons between the 4-step VCM, 50-step teacher T2V, 4-step T2V-Turbo and 8-step T2V-Turbo generations. **Left**: (VC2), **Right**: (MS).

We hire human annotators from Amazon Mechanical Turk to compare videos generated from different models given the same prompt. For each comparison, the annotators need to answer three questions: Q1) Which video is more visually appealing? Q2) Which video better fits the text description? Q3) Which video do you prefer given the prompt? Appendix D includes additional details about how we set up the human evaluations.

Figure 3 provides the full human evaluation results. We also qualitatively compare different methods in Figure 4. Due to limited space, we include additional qualitative comparison results in Appendix F. Notably, the 4-step generations from our T2V-Turbo are favored by humans over the 50-step generation from their teacher T2V model, representing a 12.5 times inference acceleration with improving performance. By increasing the inference steps to 8, we can further improve the visual quality and text-video alignment of videos generated from our T2V-Turbo, reflected by the fact that our 8-step generations are more likely to be favored by the human compared to our 4-step generations in terms of all 3 evaluated metrics. Additionally, our T2V-Turbo significantly outperforms its baseline VCM, demonstrating the effectiveness of our methods, which incorporate a mixture of reward feedback into the model training.

Table 2: Ablation studies on the effectiveness of $\mathcal{R}_{img}$ and $\mathcal{R}_{vid}$. We bold the highest score for each dimension for methods with the same teacher model. While incorporating feedback from $\mathcal{R}_{img}$ is effective at improving both Quality Score and Semantic Score, integrating reward feedback from $\mathcal{R}_{vid}$ can further improve the semantic score.

| Models | Total Score | Quality Score | Subject Consist. | BG Consist. | Temporal Flicker. | Motion Smooth. | Aesthetic Quality | Dynamic Degree | Image Quality |
|---|---|---|---|---|---|---|---|---|---|
| VCM (MS) | 75.84 | 78.80 | 93.06 | 97.30 | **98.51** | **98.00** | 48.99 | 46.11 | 61.98 |
| VCM (MS) + $\mathcal{R}_{vid}$ | 77.28 | 78.76 | 93.24 | 97.67 | 98.49 | 97.27 | 51.70 | 55.00 | 56.40 |
| VCM (MS) + $\mathcal{R}_{img}$ | 79.51 | 81.81 | **97.64** | **99.59** | 98.46 | 95.83 | **64.69** | 38.33 | **68.86** |
| Our T2V-Turbo (MS) | **80.62** | **82.15** | 94.82 | 98.71 | 97.99 | 95.64 | 60.04 | **66.39** | 68.09 |
| VCM (VC2) | 73.97 | 78.54 | 94.02 | 96.05 | **99.06** | **98.84** | 54.56 | 42.50 | 52.72 |
| VCM (VC2) + $\mathcal{R}_{vid}$ | 77.57 | 80.08 | 95.46 | 96.69 | 98.78 | 98.79 | 58.66 | 25.00 | 65.75 |
| VCM (VC2) + $\mathcal{R}_{img}$ | 80.42 | **82.59** | **96.52** | **97.31** | 97.50 | 97.29 | 63.08 | 47.50 | **72.91** |
| Our T2V-Turbo (VC2) | **81.01** | 82.57 | 96.28 | 97.02 | 97.48 | 97.34 | 63.04 | **49.17** | 72.49 |

| Models | Semantic Score | Object Class | Multiple Objects | Human Action | Color | Spatial Relation. | Scene | Appear. Style | Temporal Style | Overall Consist. |
|---|---|---|---|---|---|---|---|---|---|---|
| VCM (MS) | 63.98 | 83.18 | 24.85 | 87.20 | 85.72 | 31.57 | 42.44 | 23.20 | 23.30 | 24.18 |
| VCM (MS) + $\mathcal{R}_{vid}$ | 71.35 | 91.14 | 45.64 | 94.60 | 86.97 | 39.74 | **48.55** | 22.90 | 25.91 | 26.81 |
| VCM (MS) + $\mathcal{R}_{img}$ | 70.32 | 91.30 | 56.10 | 94.80 | 76.45 | **46.04** | 47.56 | 21.30 | 23.47 | 25.98 |
| Our T2V-Turbo (MS) | **74.47** | **93.34** | **58.63** | **95.80** | **89.67** | 45.74 | 48.47 | **23.23** | **25.92** | **27.51** |
| VCM (VC2) | 55.66 | 63.97 | 10.81 | 82.60 | 79.12 | 23.06 | 18.49 | 25.29 | 22.31 | 25.15 |
| VCM (VC2) + $\mathcal{R}_{vid}$ | 67.55 | 87.77 | 30.38 | 93.00 | 86.90 | 28.81 | 39.07 | **25.75** | 24.65 | 27.57 |
| VCM (VC2) + $\mathcal{R}_{img}$ | 71.70 | 93.13 | 46.20 | 95.00 | 84.12 | 37.78 | 51.34 | 23.65 | 24.62 | 27.75 |
| Our T2V-Turbo (VC2) | **74.76** | **93.96** | **54.65** | **95.20** | **89.90** | **38.67** | **55.58** | 24.42 | **25.51** | **28.16** |

Table 3: Effect of different choices of $\mathcal{R}_{vid}$. Our T2V-Turbo can always outperform VCM + $\mathcal{R}_{img}$ with either ViCLIP or InternVid2 S2 as $\mathcal{R}_{vid}$. Table 5 in Appendix E includes further details.

| | T2V-Turbo (VC2) $\mathcal{R}_{vid}$ = ViCLIP | T2V-Turbo (VC2) $\mathcal{R}_{vid}$ = InternVid S2 | T2V-Turbo (MS) $\mathcal{R}_{vid}$ = ViCLIP | T2V-Turbo (MS) $\mathcal{R}_{vid}$ = InternVid S2 |
|---|---|---|---|---|
| Total Score | 80.92 | **81.01** | 80.62 | 79.90 |
| Quality Score | **82.77** | 82.57 | 82.15 | 82.27 |
| Semantic Score | 73.52 | **74.76** | 74.47 | 70.41 |

## 4.3 Ablation Studies

We are interested in the effectiveness of each RM, and especially in the impact of the video-text RM $\mathcal{R}_{vid}$. Therefore, we ablate $\mathcal{R}_{img}$ and $\mathcal{R}_{vid}$ and experiment with different choices of $\mathcal{R}_{vid}$. In Appendix E, we further experiment with different choices of $\mathcal{R}_{img}$.

**Ablating RMs $\mathcal{R}_{img}$ and $\mathcal{R}_{vid}$.** Recall that the training of our T2V-Turbo incorporate reward feedback from both $\mathcal{R}_{img}$ and $\mathcal{R}_{vid}$. To demonstrate the effectiveness of each individual RM, we perform ablation study by training VCM (VC2) + $\mathcal{R}_{vid}$ and VCM (VC2) + $\mathcal{R}_{img}$, which only incorporate feedback from $\mathcal{R}_{vid}$ and $\mathcal{R}_{img}$, respectively. Again, we evaluate the 4-step generations from different methods on VBench. Results in Table 2 show that incorporating feedback from either $\mathcal{R}_{img}$ or $\mathcal{R}_{vid}$ leads to performance improvement over the baseline VCM. Notably, optimizing $\mathcal{R}_{img}$ alone can already lead to substantial performance gains, while incorporating feedback from $\mathcal{R}_{vid}$ can further improve the Semantic Score on VBench, leading to better text-video alignment. In Appendix H, we qualitatively compare the videos generated by our T2V-Turbo and VCM + $\mathcal{R}_{img}$, corroborating the effectiveness of our mixture of RMs design.

**Effect of different choices of $\mathcal{R}_{vid}$.** We investigate the impact of different choices of $\mathcal{R}_{vid}$ by training T2V-Turbo (VC2) and T2V-Turbo (MS) by setting $\mathcal{R}_{vid}$ as ViCLIP [Wang et al., 2023d] and the second stage model of Intervideo2 (InternVid2 S2). In terms of model architecture, ViCLIP employs the CLIP [Radford et al., 2021] text encoder while InternVid2 S2 leverages the BERT-large [Kenton and Toutanova, 2019] text encoder. Additionally, InternVid2 S2 outperforms ViCLIP in several zero-shot video-text retrieval tasks. As shown in Table 3, T2V-Turbo (VC2) can achieve decent performance on VBench when integrating feedback from either ViCLIP or InternVid2 S2. Conversely, T2V-Turbo (MS) performs better with ViCLIP [Wang et al., 2023d]. Nevertheless, with InternVid2 S2, our T2V-Turbo (MS) still surpasses VCM (MS) + $\mathcal{R}_{img}$.

# 5 Related Work

**Diffusion-based T2V Models**. Many diffusion-based T2V models rely on large-scale image datasets for training [Ho et al., 2022a, Wang et al., 2023c, Chen et al., 2023] or inherit weights from pre-trained text-to-image (T2I) models [Zhang et al., 2023, Blattmann et al., 2023, Khachatryan et al., 2023]. The scale of text-image datasets [Schuhmann et al., 2022] is usually more than ten times the scale of open-sourced video-text datasets [Bain et al., 2021, Wang et al., 2023d] and with higher spatial resolution and diversity [Wang et al., 2023c]. For example, Imagen Video [Ho et al., 2022b] discovers that joint training on a mix of image and video datasets improves the overall visual quality and enables the generation of videos in novel styles. Models trained with WebVid-10M [Bain et al., 2021] like ModelScopeT2V [Wang et al., 2023c] or VideoCrafter [Chen et al., 2023] also treat images as a single-frame video, and use them to improve video qualities. LaVie [Wang et al., 2023b] initialize the training with WebVid-10M and LAION-5B and then continue the training with a curated internal dataset of 23M videos. To overcome the data scarcity of high-quality videos, VideoCrafter2 [Chen et al., 2024] proposes to disentangle motion from appearance at the data level so that it can be trained on high-quality images and low-quality videos. The data limitation of high-quality videos and aligned, accurate video captions has been a longstanding bottleneck of current T2V models. In this paper, we propose to combat this challenge by leveraging reward feedback from a mixture of RMs.

**Accelerating inference of Diffusion Models**. Various methods have been proposed to accelerate the sampling process of a DM, including advanced numerical ODE solvers [Song et al., 2020b, Lu et al., 2022a,b, Zheng et al., 2022, Dockhorn et al., 2022, Jolicoeur-Martineau et al., 2021] and distillation techniques [Luhman and Luhman, 2021, Salimans and Ho, 2021, Meng et al., 2023, Zheng et al., 2023]. Recently, Consistency Model [Song et al., 2023, Luo et al., 2023a] is proposed to facilitate fast inference by learning a consistency function to map any point at the ODE trajectory to the origin. Li et al. [2024] proposes to augment consistency distillation with an objective to optimize image-text RM to achieve fast and high-quality image generation. Our work extends it for T2V generation, incorporating reward feedback from both an image-text RM and a video-text RM.

**Vision-and-language Reward Models.** There have been various open-sourced image-text RMs that are trained to mirror human preferences given a text-image pair, including HPS [Wu et al., 2023b,a], ImageReward [Xu et al., 2024a], and PickScore [Kirstain et al., 2024], which are obtained by finetuning a image-text foundation model such as CLIP [Radford et al., 2021] and BLIP [Li et al., 2022], on human preference data. However, to the best of our knowledge, no video-text RMs, e.g., T2VScore [Wu et al., 2024], that mirrors human preference on a text-video pair has been released to the public. In this paper, we choose HPSv2.1 as our image-text RM and directly employ the video foundation models ViCLIP [Wang et al., 2023d] and InterVid S2 [Wang et al., 2024] that are trained for general video-text understanding as our video-text RM. Empirically, we show that incorporating feedback from these RMs can improve the performance of our `T2V-Turbo`.

**Learning from Human/AI Feedback** has been proven as an effective way to align the output from a generative model with human preference [Leike et al., 2018, Ziegler et al., 2019, Ouyang et al., 2022, Stiennon et al., 2020, Rafailov et al., 2024, Xu et al., 2024b]. In the field of image generation, various methods have been proposed to align a text-to-image model with human preference, including RL [Sutton and Barto, 2018, Li et al., 2020, 2023a] based methods [Fan et al., 2024, Prabhudesai et al., 2023, Zhang et al., 2024] and backpropagation-based reward finetuning methods [Clark et al., 2023, Xu et al., 2024a, Prabhudesai et al., 2023]. Recently, InstructVideo [Yuan et al., 2023] extends the reward-finetuning methods to optimize a T2V model. However, it still employs an image-text RM to provide reward feedback without considering the transition dynamic of the generated video. In contrast, our work incorporates reward feedback from both an image-text and video-text RM, providing comprehensive feedback to our `T2V-Turbo`.

# 6 Conclusion and Limitations

In this paper, we propose `T2V-Turbo`, achieving both fast and high-quality T2V generation by breaking the quality bottleneck of a VCM. Specifically, we integrate mixed reward feedback into the VCD process of a teacher T2V model. Empirically, we illustrate the applicability of our methods by distilling `T2V-Turbo` (VC2) and `T2V-Turbo` (MS) from VideoCrafter2 [Chen et al., 2024] and ModelScopeT2V [Wang et al., 2023c], respectively. Remarkably, the 4-step generations from both our `T2V-Turbo` outperform SOTA methods on VBench [Huang et al., 2024], even surpassing their

teacher T2V models and proprietary systems including Gen-2 [Esser et al., 2023] and Pika [Pika Labs, 2023]. Our human evaluation further corroborates the results, showing the 4-step generations from our `T2V-Turbo` are favored by humans over the 50-step DDIM samples from their teacher, which represents over ten-fold inference acceleration with quality improvement.

While our `T2V-Turbo` marks a critical advancement in efficient T2V synthesis, it is important to recognize certain limitations. Our approach utilizes a mixture of RMs, including a video-text RM $\mathcal{R}_{\text{vid}}$. Due to the lack of an open-sourced video-text RM trained to reflect human preferences on video-text pairs, we instead use video foundation models such as ViCLIP [Wang et al., 2023d] and InternVid S2 [Wang et al., 2024] as our $\mathcal{R}_{\text{vid}}$. Although incorporating feedback from these models has enhanced our `T2V-Turbo`'s performance, future research should explore the use of a more advanced $\mathcal{R}_{\text{vid}}$ for training feedback, which could lead to further performance improvements.

## Acknowledgement

The work was funded by an unrestricted gift from Google, and we are grateful for their generous sponsorship. The views and conclusions contained in this document are those of the authors and should not be interpreted as representing the sponsors' official policy, expressed or inferred.

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

# Appendix

## A    Experiment and Hyperparameter (HP) Details

When performing qualitative comparisons between different methods, we ensure to use the same random seed for head-to-head video comparisons.

As mentioned in Sec. 4, we train `T2V-Turbo` (VC2) and `T2V-Turbo` (MS) by distilling from the teacher diffusion-based T2V models VideoCrafter2 [Chen et al., 2024] and the less advanced ModelScopeT2V [Wang et al., 2023c], respectively. Specifically, VideoCrafter2 supports video FPS as an input and generates videos at the resolution of 512x320. For simplicity, we always set FPS to 16 when distilling our `T2V-Turbo` (VC2) from VideoCrafter2. On the other hand, ModelScopeT2V always generates video at 8 FPS at a resolution of 256x256.

We conduct our training with the WebVid10M [Bain et al., 2021] datasets. Note that both teacher T2V models are also trained with WebVid10M. We train our models for 10K gradient steps with 6 - 8 NVIDIA A100 GPUs without gradient accumulation and set the batch size to 1 for each GPU device. That is, we load 1 video with 16 frames. At each training iteration, we always sample 16 frames from the input video. We employ HPSv2.1 [Wu et al., 2023a] as our image-text RM $\mathcal{R}_{\text{img}}$. When distilling from VideoCrafter2, we utilize the 2nd Stage model of InternVideo2 [Wang et al., 2024] as our video-text RM $\mathcal{R}_{\text{vid}}$. When distilling from ModelScopeT2V, we set $\mathcal{R}_{\text{vid}}$ to be ViCLIP [Wang et al., 2023d]. To optimize $J_{\text{img}}$ (8), we randomly sample 6 frames from the video by setting $M = 6$. For the hyperparameters, we set learning rate $1e - 5$ and guidance scale range $[\omega_{\text{min}}, \omega_{\text{max}}] = [5, 15]$. We use DDIM [Song et al., 2020b] as our ODE solver $\Psi$ and set the skipping step $k = 20$. For `T2V-Turbo` (VC2), we set $\beta_{\text{img}} = 1$ and $\beta_{\text{vid}} = 2$. For `T2V-Turbo` (MS), we set $\beta_{\text{img}} = 2$ and $\beta_{\text{vid}} = 3$.

As mentioned in Sec. 2, we employ the DDIM [Song et al., 2020b] ODE solver $\Psi_{\text{DDIM}}$ by following the practice of Luo et al. [2023a]. Its formula from $t_{n+k}$ to $t_n$ is given below

$$\Psi_{\text{DDIM}}\left(\boldsymbol{z}_{t_{n+k}}, t_{n+k}, t_n, \boldsymbol{c}\right) = \underbrace{\frac{\alpha_{t_n}}{\alpha_{t_{n+k}}} \boldsymbol{z}_{t_{n+k}} - \beta_{t_n}\left(\frac{\beta_{t_{n+k}} \cdot \alpha_{t_n}}{\alpha_{t_{n+k}} \cdot \beta_{t_n}} - 1\right) \hat{\boldsymbol{\epsilon}}_{\psi}\left(\boldsymbol{z}_{t_{n+k}}, \boldsymbol{c}, t_{n+k}\right)}_{\text{DDIM Estimated } \boldsymbol{z}_{t_n}} - \boldsymbol{z}_{t_{n+k}}$$

(11)

where $\hat{\boldsymbol{\epsilon}}_{\psi}$ denotes the noise prediction model from the teacher T2V model. We refer interested readers to the original LCM paper [Luo et al., 2023a] for further details.

# B Psudoe-codes for Training our `T2V-Turbo`

We include the pseudo-codes for training our `T2V-Turbo` in Algorithm 1. We use the red color to highlight the difference from the standard (latent) consistency distillation [Luo et al., 2023a, Song et al., 2023].

---

**Algorithm 1** `T2V-Turbo` Training Pipeline

---

**Require:** text-video dataset $\mathcal{D}$, initial model parameter $\theta$, learning rate $\eta$, ODE solver $\Psi$, distance metric $d$, EMA rate $\mu$, noise schedule $\alpha(t), \beta(t)$, guidance scale $[\omega_{\min}, \omega_{\max}]$, skipping interval $k$, VAE encoder $\mathcal{E}$, decoder $\mathcal{D}$, image-text RM $\mathcal{R}_{\text{img}}$, video-text RM $\mathcal{R}_{\text{vid}}$, reward scale $\beta_{\text{img}}$ and $\beta_{\text{vid}}$.

Encoding training data into latent space: $\mathcal{D}_z = \{(\boldsymbol{z}, \boldsymbol{c}) \mid \boldsymbol{z} = E(\boldsymbol{x}), (\boldsymbol{x}, \boldsymbol{c}) \in \mathcal{D}\}$

$\theta^- \leftarrow \theta$

**repeat**

    Sample $(\boldsymbol{z}, \boldsymbol{c}) \sim \mathcal{D}_z, n \sim \mathcal{U}[1, N-k]$ and $\omega \sim [\omega_{\min}, \omega_{\max}]$

    Sample $\boldsymbol{z}_{t_{n+k}} \sim \mathcal{N}\left(\alpha\left(t_{n+k}\right) \boldsymbol{z}; \sigma^2\left(t_{n+k}\right) \mathbf{I}\right)$

    $\hat{\boldsymbol{z}}_{t_n}^{\Psi, \omega} \leftarrow \boldsymbol{z}_{t_{n+k}} + (1+\omega)\Psi\left(\boldsymbol{z}_{t_{n+k}}, t_{n+k}, t_n, \boldsymbol{c}\right) - \omega\Psi\left(\boldsymbol{z}_{t_{n+k}}, t_{n+k}, t_n, \varnothing\right)$

    $\hat{\mathbf{x}}_0 = \mathcal{D}\left(\boldsymbol{f}_\theta\left(\mathbf{z}_{t_{n+k}}, \omega, \mathbf{c}, t_{n+k}\right)\right)$

    $J_{\text{img}}(\theta) = \mathbb{E}_{\hat{\mathbf{x}}_0, \mathbf{c}}\left[\sum_{m=1}^M \mathcal{R}_{\text{img}}\left(\hat{\mathbf{x}}_0^m, \mathbf{c}\right)\right]$

    $J_{\text{vid}}(\theta) = \mathbb{E}_{\hat{\mathbf{x}}_0, \mathbf{c}}\left[\mathcal{R}_{\text{vid}}\left(\hat{\mathbf{x}}_0, \mathbf{c}\right)\right]$

    $L_{\text{CD}} = d\left(\boldsymbol{f}_\theta\left(\boldsymbol{z}_{t_{n+k}}, \omega, \boldsymbol{c}, t_{n+k}\right), \boldsymbol{f}_{\theta^-}\left(\hat{\boldsymbol{z}}_{t_n}^{\Psi, \omega}, \omega, \boldsymbol{c}, t_n\right)\right)$

    $\mathcal{L}\left(\theta, \theta^-; \Psi\right) \leftarrow L_{\text{CD}} - \beta_{\text{img}} J_{\text{img}}(\theta) - \beta_{\text{vid}} J_{\text{vid}}(\theta)$

    $\theta \leftarrow \theta - \eta \nabla_\theta \mathcal{L}\left(\theta, \theta^-\right)$

    $\theta^- \leftarrow \texttt{stop\_grad}\left(\mu\theta^- + (1-\mu)\theta\right)$

**until** convergence

---

# C   Further Details about VBench

We provide a brief introduction of the metrics included in VBench [Huang et al., 2024] followed by introducing the derivation rules for the **Quality Score**, **Semantic Score** and **Total Score**. We refer interested readers to read the VBench paper for further details.

The following metrics are used to construct the **Quality Score**.

- **Subject Consistency** (Subject Consist.) is calculated by the DINO [Caron et al., 2021] feature similarity across video frames.
- **Background Consistency** (BG Consist.) is calculated by CLIP [Radford et al., 2021] feature similarity across video frames.
- **Temporal Flickering** (Temporal Flicker.) is computed by the mean absolute difference across video frames.
- **Motion Smoothness** (Motion Smooth.) is evaluated by motion priors in the video frame interpolation model [Li et al., 2023b].
- **Aesthetic Quality** is calculated by mean of aesthetic scores evalauted by the LAION aesthetic predictor [Schuhmann et al., 2022].
- **Dynamic Degree** is calculated using RAFT [Teed and Deng, 2020].
- **Image Quality** is evaluated by the MUSIQ [Ke et al., 2021] image quality predictor.

**Quality Score** is calculated as the weighted sum of the normalized scores of each metric mentioned above. The weight for all metrics is 1, except for **Dynamic Degree**, which has a weight of 0.5.

The following metrics are used to construct the **Semantic Score**.

- **Object Class** is calculated by detecting the success rate of generating the object specified by the user using GRiT [Wu et al., 2022].
- **Multiple Object** is calculated by detecting the success rate of generating all objects specified in the prompt using GRiT [Wu et al., 2022].
- **Human Action** is evaluated by the UMT model [Li et al., 2023c].
- **Color** is calculated by comparing the color caption generated by GRiT [Wu et al., 2022] against the expected color.
- **Spatial Relationship** (Spatial Relation.) is calculated by a rule-based method similar to [Huang et al., 2023a].
- **Scene** is calculated by comparing the video captions generated by Tag2Text [Huang et al., 2023b] against the scene descriptions in the prompt.
- **Appearance Style** (Appear Style.) is calculated by using ViCLIP [Wang et al., 2023d] to compare the video feature and the style description in the user prompt.
- **Temporal Style** is calculated based on the similarity between the video feature and the style descrption feature provided by ViCLIP [Wang et al., 2023d].
- **Overall Consistency** (Overall Consist.) is calculated based on the similarity between the video feature and the entire text prompt feature provided by ViCLIP [Wang et al., 2023d]. ViCLIP [Wang et al., 2023d]

**Semantic Score** is simply calculated as the mean of the normalized scores of each metric mentioned above. And the **Total Score** is the weighted sum of **Quality Score** and **Semantic Score**, which is given by

$$\textbf{Total Score} = \frac{4 \cdot \textbf{Quality Score} + \textbf{Total Score}}{5} \tag{12}$$

Table 4: **Automatic Evaluation on VBench** [Huang et al., 2024]. We compare our `T2V-Turbo` (VC2) and `T2V-Turbo` (MS) with baseline methods across the 16 VBench dimensions. A higher score indicates better performance for a particular dimension. We bold the best results for each dimension and underline the second-best result. **Quality Score** is calculated with the 7 dimensions from the top table. **Semantic Score** is calculated with the 9 dimensions from the bottom table. **Total Score** a weighted sum of **Quality Score** and **Semantic Score**. Both our `T2V-Turbo` (VC2) and `T2V-Turbo` (MS) **surpass all baseline methods with 4 inference steps** in terms of Total Score, including the proprietary systems Gen-2 and Pika.

| Models | Total Score | Quality Score | Subject Consist. | BG Consist. | Temporal Flicker. | Motion Smooth. | Aesthetic Quality | Dynamic Degree | Image Quality |
|---|---|---|---|---|---|---|---|---|---|
| CogVideo | 67.01 | 72.06 | 92.19 | 96.20 | 97.64 | 96.47 | 38.18 | 42.22 | 41.03 |
| VideoCrafter0.9 | 73.02 | 74.91 | 86.24 | 92.88 | 97.60 | 91.79 | 44.41 | **89.72** | 57.22 |
| ModelScopeT2V | 75.75 | 78.05 | 89.87 | 95.29 | 98.28 | 95.79 | 52.06 | 66.39 | 58.57 |
| Open-Sora | 75.91 | 78.82 | 92.09 | 97.39 | 98.41 | 95.61 | 57.76 | 48.61 | 61.51 |
| LaVie | 77.08 | 78.78 | 91.41 | 97.47 | 98.30 | 96.38 | 54.94 | 49.72 | 61.90 |
| LaVie-Interpolation | 77.12 | 79.07 | 92.00 | 97.33 | 98.78 | 97.82 | 54.00 | 46.11 | 59.78 |
| Show-1 | 78.93 | 80.42 | 95.53 | 98.02 | 99.12 | 98.24 | 57.35 | 44.44 | 58.66 |
| VideoCrafter1 | 79.72 | 81.59 | 95.10 | 98.04 | 98.93 | 95.67 | 62.67 | 55.00 | 65.46 |
| Pika | 80.40 | **82.68** | 96.76 | **98.95** | **99.77** | 99.51 | 63.15 | 37.22 | 62.33 |
| VideoCrafter2 | 80.44 | 82.20 | 96.85 | 98.22 | 98.41 | 97.73 | 63.13 | 42.50 | 67.22 |
| Gen-2 | 80.58 | 82.47 | **97.61** | 97.61 | 99.56 | **99.58** | **66.96** | 18.89 | 67.42 |
| VCM (MS) | 75.84 | 78.80 | 93.06 | 97.30 | 98.51 | 98.00 | 48.99 | 46.11 | 61.98 |
| Our T2V-Turbo (MS) | 80.62 | 82.15 | 94.82 | 98.71 | 97.99 | 95.64 | 60.04 | 66.39 | 68.09 |
| VCM (VC2) | 73.97 | 78.54 | 94.02 | 96.05 | 99.06 | 98.84 | 54.56 | 42.50 | 52.72 |
| Our T2V-Turbo (VC2) | **81.01** | 82.57 | 96.28 | 97.02 | 97.48 | 97.34 | 63.04 | 49.17 | **72.49** |

| Models | Semantic Score | Object Class | Multiple Objects | Human Action | Color | Spatial Relation. | Scene | Appear. Style | Temporal Style | Overall Consist. |
|---|---|---|---|---|---|---|---|---|---|---|
| CogVideo | 46.83 | 73.40 | 18.11 | 78.20 | 79.57 | 18.24 | 28.24 | 22.01 | 7.80 | 7.70 |
| VideoCrafter0.9 | 65.46 | 87.34 | 25.93 | 93.00 | 78.84 | 36.74 | 43.36 | 21.57 | 25.42 | 25.21 |
| ModelScopeT2V | 66.54 | 82.25 | 38.98 | 92.40 | 81.72 | 33.68 | 39.26 | 23.39 | 25.37 | 25.67 |
| Open-Sora | 64.28 | 74.98 | 33.64 | 85.00 | 78.15 | 43.95 | 37.33 | 21.58 | 25.46 | 26.18 |
| LaVie | 70.31 | 91.82 | 33.32 | **96.80** | 86.39 | 34.09 | 52.69 | 23.56 | 25.93 | 26.41 |
| LaVie-Interpolation | 69.31 | 90.68 | 30.93 | 95.80 | 85.69 | 30.06 | 52.62 | 23.53 | **26.01** | 26.51 |
| Show-1 | 72.98 | 93.07 | 45.47 | 95.60 | 86.35 | 53.50 | 47.03 | 23.06 | 25.28 | 27.46 |
| VideoCrafter1 | 72.22 | 78.18 | 45.66 | 91.60 | **93.32** | 58.86 | 43.75 | 24.41 | 25.54 | 26.76 |
| Pika | 71.26 | 87.45 | 46.69 | 88.00 | 85.31 | 65.65 | 44.80 | 21.89 | 24.44 | 25.47 |
| VideoCrafter2 | 73.42 | 92.55 | 40.66 | 95.00 | 92.92 | 35.86 | 55.29 | 25.13 | 25.84 | **28.23** |
| Gen-2 | 73.03 | 90.92 | 55.47 | 89.20 | 89.49 | **66.91** | 48.91 | 19.34 | 24.12 | 26.17 |
| VCM (MS) | 63.98 | 83.18 | 24.85 | 87.20 | 85.72 | 31.57 | 42.44 | 23.20 | 23.30 | 24.18 |
| Our T2V-Turbo (MS) | 74.47 | 93.34 | **58.63** | 95.80 | 89.67 | 45.74 | 48.47 | 23.23 | 25.92 | 27.51 |
| VCM (VC2) | 55.66 | 63.97 | 10.81 | 82.60 | 79.12 | 23.06 | 18.49 | **25.29** | 22.31 | 25.15 |
| Our T2V-Turbo (VC2) | **74.76** | **93.96** | 54.65 | 95.20 | 89.90 | 38.67 | **55.58** | 24.42 | 25.51 | 28.16 |

## D    Human Evaluation Details

Figure 5 shows the user interface displayed to the labelers when conducting our human evaluations. Each method generate videos of 16 frames using the 700 prompts from EvalCrafter [Liu et al., 2023]. For our `T2V-Turbo` (VC2), we collect its 4-step and 8-step generations and compare them with the 50-step DDIM samples from its teacher VideoCrafter2. For our `T2V-Turbo` (MS), we collect its 2-step and 4-step generations and compare them with the 50-step DDIM samples from its teacher ModelScopeT2V. We also compare the 4-step generations between our `T2V-Turbo` their baseline VCM, demonstrating the significant quality improvement of our methods.

As mentioned in Sec. 4.2, we hire labelers from Amazon Mechanical Turk platform and form the video comparison task as many batches of HITs. Specifically, we choose labelers from English-speaking countries, including AU, CA, NZ, GB, and the US. Each task needs around 30 seconds to complete, and we pay each submitted HIT with 0.2 US dollars. Therefore, the hourly payment is about 24 US dollars.

We note that the data annotation part of our project is classified as exempt by Human Subject Committee via IRB protocols.

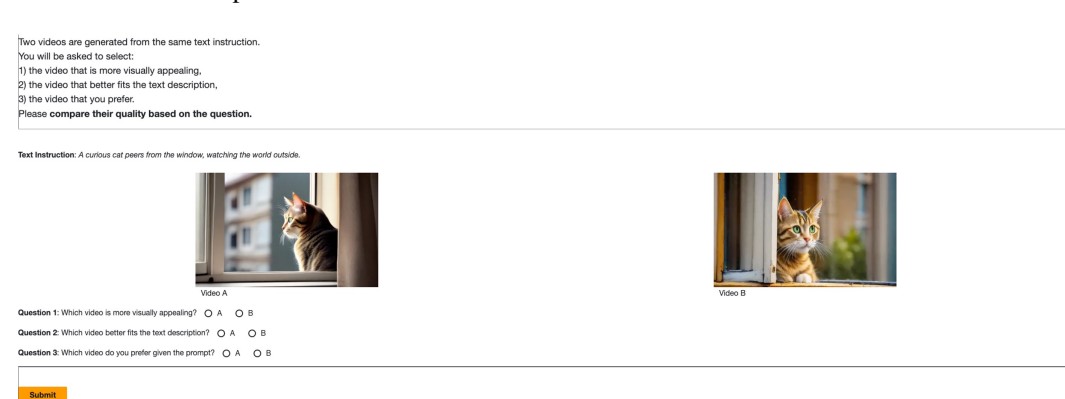

Figure 5: User interface of our human evaluation experiments.

# E  Additional Ablation Studies

In this section, we provide the full ablation results performed in Table 3, which can be found in Table 5. We further examine the effect of different choices of $\mathcal{R}_{\mathrm{img}}$. In the initial stage of our project, we train VCM (VC2) + $\mathcal{R}_{\mathrm{img}}$ with several different image-text RMs, including HPSv2.1 [Wu et al., 2023a], PickScore [Kirstain et al., 2024], and ImageReward [Xu et al., 2024a]. We collect the 4-step generations from each method and qualitatively compare them with the 4-step generation from the baseline VCM. As shown in Figure 6, incorporating reward feedback from any of these $\mathcal{R}_{\mathrm{img}}$ leads to quality improvement over the baseline VCM (VC2). It is worth noting that HPSv2.1 and PickScore are fine-tuned from CLIP with human preference data. Therefore, learning from CLIP might also lead to better performance than the baseline VCM.

Table 5: Effect of different choices of $\mathcal{R}_{\mathrm{vid}}$. T2V-Turbo (VC2) can achieve decent performance on VBench when integrating feedback from either ViCLIP or InternVid2 S2. On the other hand, T2V-Turbo (MS) achieves a better result with ViCLIP [Wang et al., 2023d].

| Models | Total Score | Quality Score | Subject Consist. | BG Consist. | Temporal Flicker. | Motion Smooth. | Aesthetic Quality | Dynamic Degree | Image Quality |
|---|---|---|---|---|---|---|---|---|---|
| T2V-Turbo (MS), $\mathcal{R}_{\mathrm{vid}}$ = ViCLIP | **80.62** | 82.15 | 94.82 | 98.71 | **97.99** | 95.64 | 60.04 | **66.39** | 68.09 |
| T2V-Turbo (MS), $\mathcal{R}_{\mathrm{vid}}$ = InternVid2 S2 | 79.90 | **82.27** | **96.68** | **99.36** | 97.74 | **95.66** | **65.30** | 52.22 | **68.23** |
| T2V-Turbo (VC2), $\mathcal{R}_{\mathrm{vid}}$ = ViCLIP | 80.92 | **82.77** | **96.93** | **97.47** | **98.03** | **97.48** | **63.38** | 43.61 | **72.94** |
| T2V-Turbo (VC2), $\mathcal{R}_{\mathrm{vid}}$ = InternVid2 S2 | **81.01** | 82.57 | 96.28 | 97.02 | 97.48 | 97.34 | 63.04 | **49.17** | 72.49 |

| Models | Semantic Score | Object Class | Multiple Objects | Human Action | Color | Spatial Relation. | Scene | Appear. Style | Temporal Style | Overall Consist. |
|---|---|---|---|---|---|---|---|---|---|---|
| T2V-Turbo (MS), $\mathcal{R}_{\mathrm{vid}}$ = ViCLIP | **74.47** | 93.34 | **58.63** | **95.80** | **89.67** | **45.74** | **48.47** | **23.23** | **25.92** | **27.51** |
| T2V-Turbo (MS), $\mathcal{R}_{\mathrm{vid}}$ = InternVid2 S2 | 70.41 | **94.05** | 48.73 | 92.60 | 81.69 | 45.41 | 48.15 | 21.45 | 23.84 | 26.24 |
| T2V-Turbo (VC2), $\mathcal{R}_{\mathrm{vid}}$ = ViCLIP | 73.52 | **94.05** | 50.52 | 94.40 | 89.85 | 36.77 | 54.17 | 23.81 | 25.34 | 28.11 |
| T2V-Turbo (VC2), $\mathcal{R}_{\mathrm{vid}}$ = InternVid2 S2 | **74.76** | 93.96 | **54.65** | **95.20** | **89.90** | **38.67** | **55.58** | **24.42** | **25.51** | **28.16** |

# F  Qualitative Results

We provide additional qualitative comparisons between our T2V-Turbo, the baseline VCM, and their teacher T2V models in Figures 7, 8, 9, and 10.

The prompts for the top two and bottom two rows in Figure 1 are given below:

- With the style of low-poly game art, A majestic, white horse gallops gracefully across a moonlit beach.
- Kung Fu Panda posing in cyberpunk, neonpunk style.

# G  Broader Impact

The ability to create highly realistic synthetic videos raises concerns about misinformation and deepfakes, which can be used to manipulate public opinion, defame individuals, or perpetrate fraud. Addressing these concerns requires robust regulatory frameworks and ethical guidelines to ensure the technology is used responsibly and for the benefit of society. **Therefore, we are committed to installing safeguard when releasing our models**. Specifically, we will require users to adhere to usage guidelines.

Despite the challenges, the impact of our T2V-Turbo is profound, offering a scalable solution that significantly enhances the accessibility and practicality of generating high-quality videos at a remarkable speed. This innovation not only broadens the potential applications in fields ranging from digital art to visual content creation but also sets a new benchmark for future research in T2V synthesis, emphasizing the importance of human-centric design in the development of generative AI technologies.

*a cat drinking beer*

VCM (VC2)

VCM (VC2)
+ HPSv2,1

VCM (VC2)
+ PickScore

VCM (VC2)
+ ImgRwd

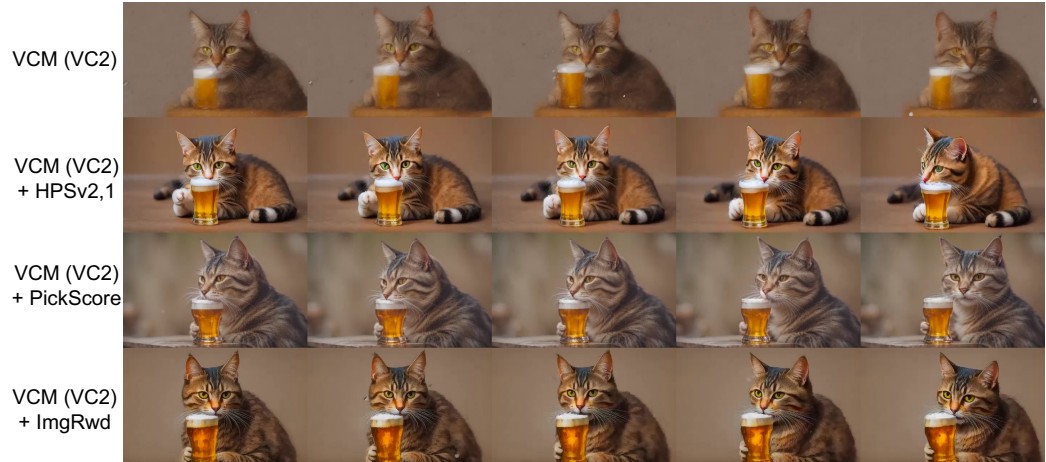

*a dog wearing vr goggles on a boat*

VCM (VC2)

VCM (VC2)
+ HPSv2,1

VCM (VC2)
+ PickScore

VCM (VC2)
+ ImgRwd

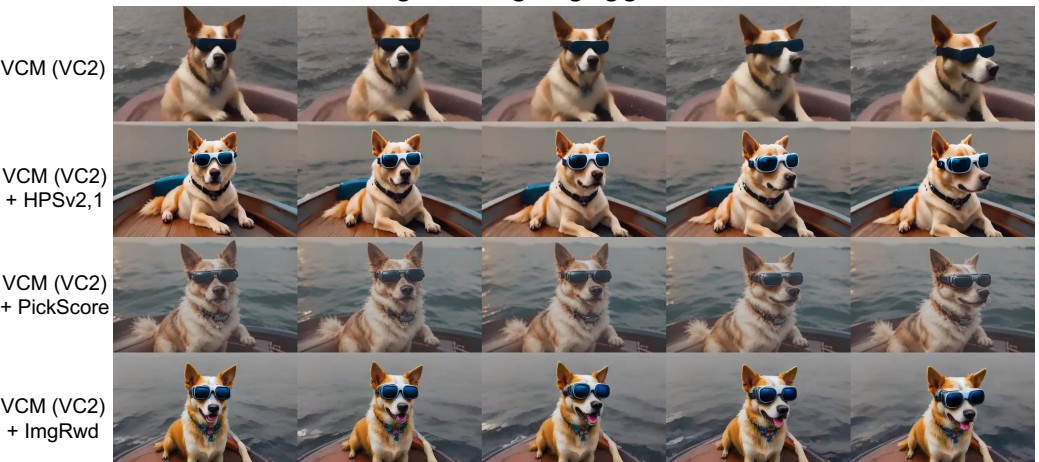

*With the style of low-poly game art, A majestic, white
horse gallops gracefully across a moonlit beach.*

VCM (VC2)

VCM (VC2)
+ HPSv2,1

VCM (VC2)
+ PickScore

VCM (VC2)
+ ImgRwd

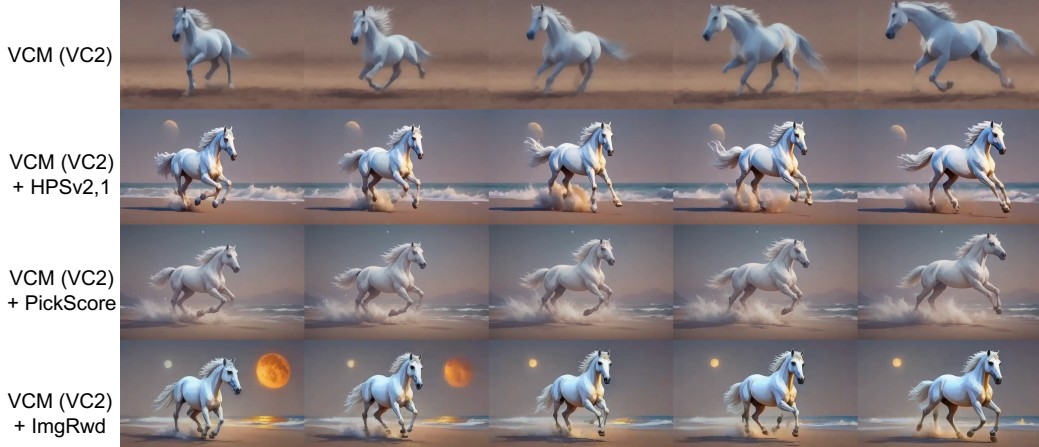

Figure 6: Ablation study on the choice of the $\mathcal{R}_{\text{img}}$. We compare the 4-step generations from each methods. The three $\mathcal{R}_{\text{img}}$ we tested can all improve the video generation quality compare to the baseline VCM (VC2).

*A wise tortoise in a tweed hat and spectacles reads a newspaper, Howard Hodgkin style*

VCM (MS)
(4-step)

VideoCrafter2
(50-step)

**T2V-Turbo
(VC2) 4-step**

**T2V-Turbo
(VC2) 8-step**

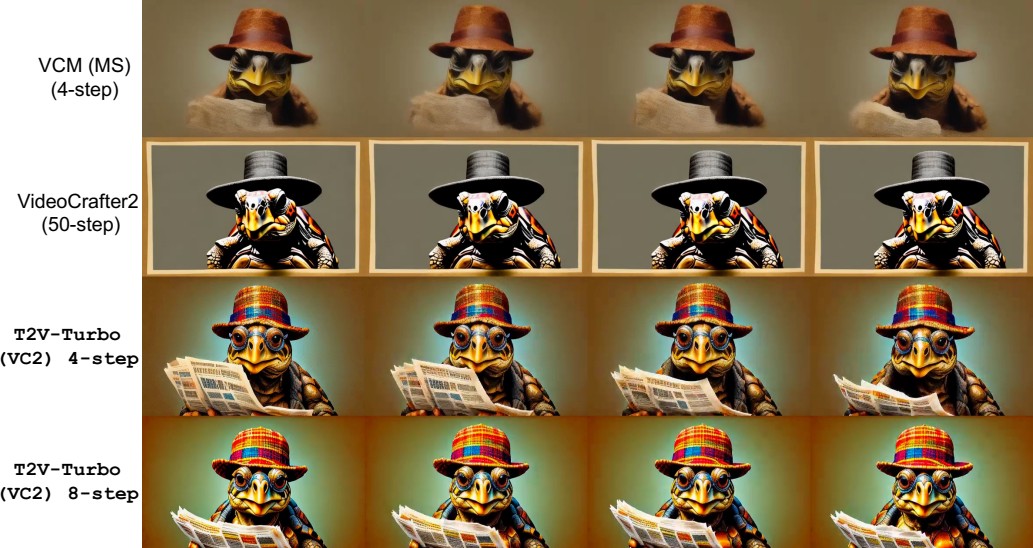

*drove viewpoint, fireworks above the Parthenon*

VCM (MS)
(4-step)

VideoCrafter2
(50-step)

**T2V-Turbo
(VC2) 4-step**

**T2V-Turbo
(VC2) 8-step**

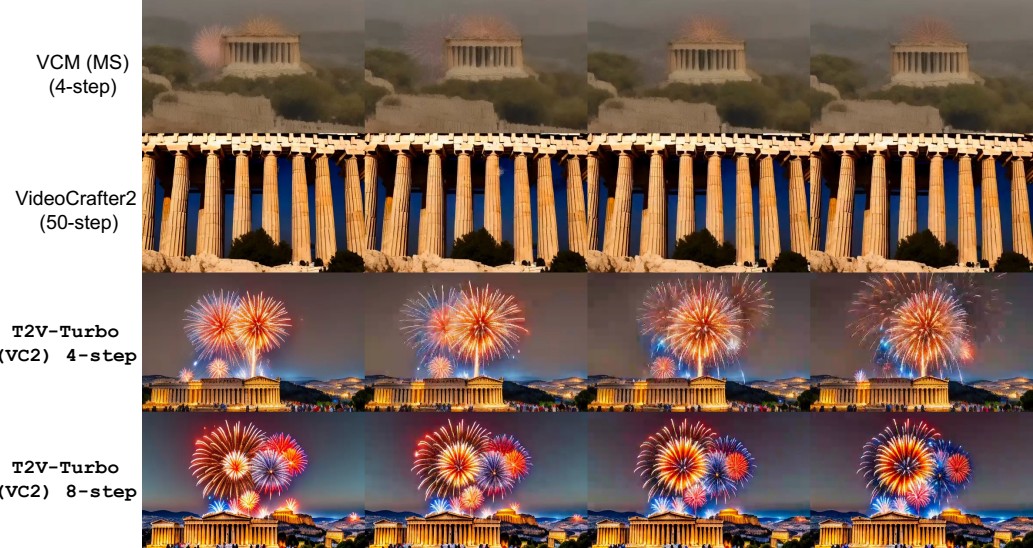

*Iron Man is walking towards the camera in the rain at night, with a lot of fog behind him. Science fiction movie, close-up*

VCM (MS)
(4-step)

VideoCrafter2
(50-step)

**T2V-Turbo
(VC2) 4-step**

**T2V-Turbo
(VC2) 8-step**

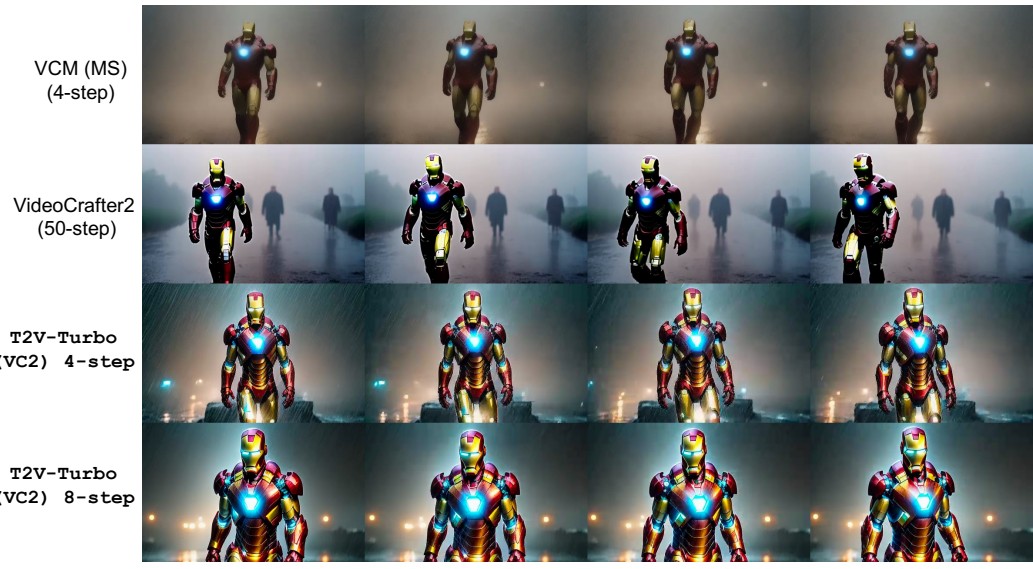

Figure 7: Additional qualitative comparison results for our T2V-Turbo (VC2).

*With the style of sketch, A sophisticated monkey in a beret and striped shirt paints in a French artist's studio.*

VCM (MS) (4-step)

VideoCrafter2 (50-step)

T2V-Turbo (VC2) 4-step

T2V-Turbo (VC2) 8-step

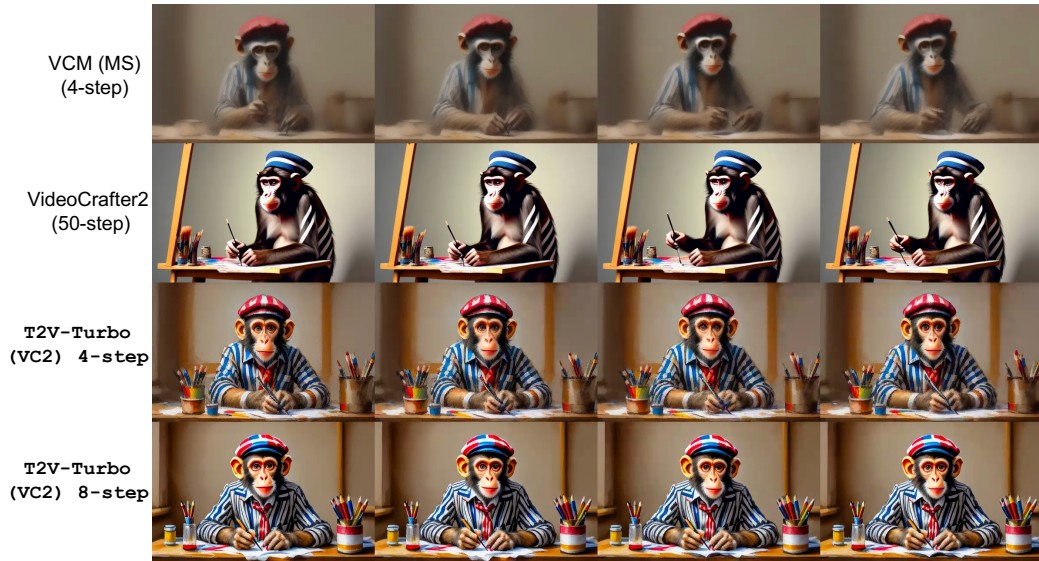

*a cyborg standing on top of a skyscraper, overseeing the city, back view, cyberpunk vibe, 2077, NYC, intricate details, 4K*

VCM (MS) (4-step)

VideoCrafter2 (50-step)

T2V-Turbo (VC2) 4-step

T2V-Turbo (VC2) 8-step

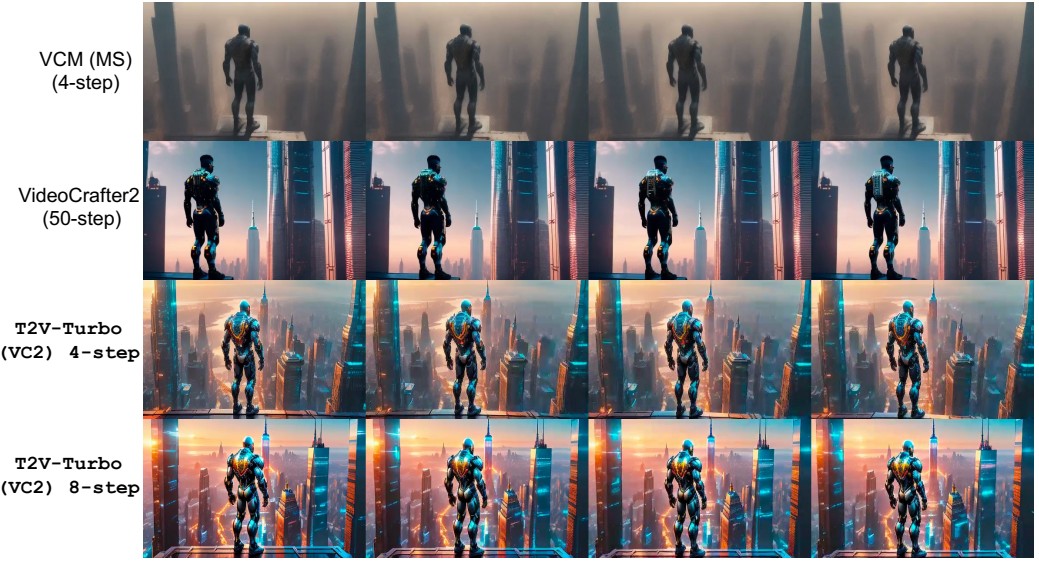

*A Egyptian tomp hieroglyphics painting of A regal lion, decked out in a jeweled crown, surveys his kingdom.*

VCM (MS) (4-step)

VideoCrafter2 (50-step)

T2V-Turbo (VC2) 4-step

T2V-Turbo (VC2) 8-step

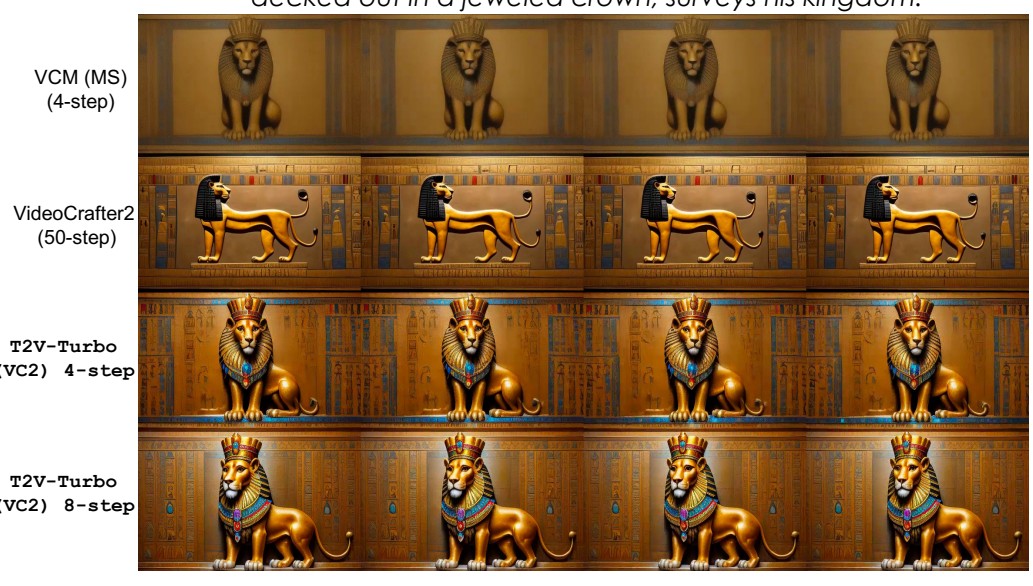

Figure 8: Additional qualitative comparison results for our T2V-Turbo (VC2).

*Macro len style, A tiny mouse in a dainty dress holds a parasol to shield from the sun.*

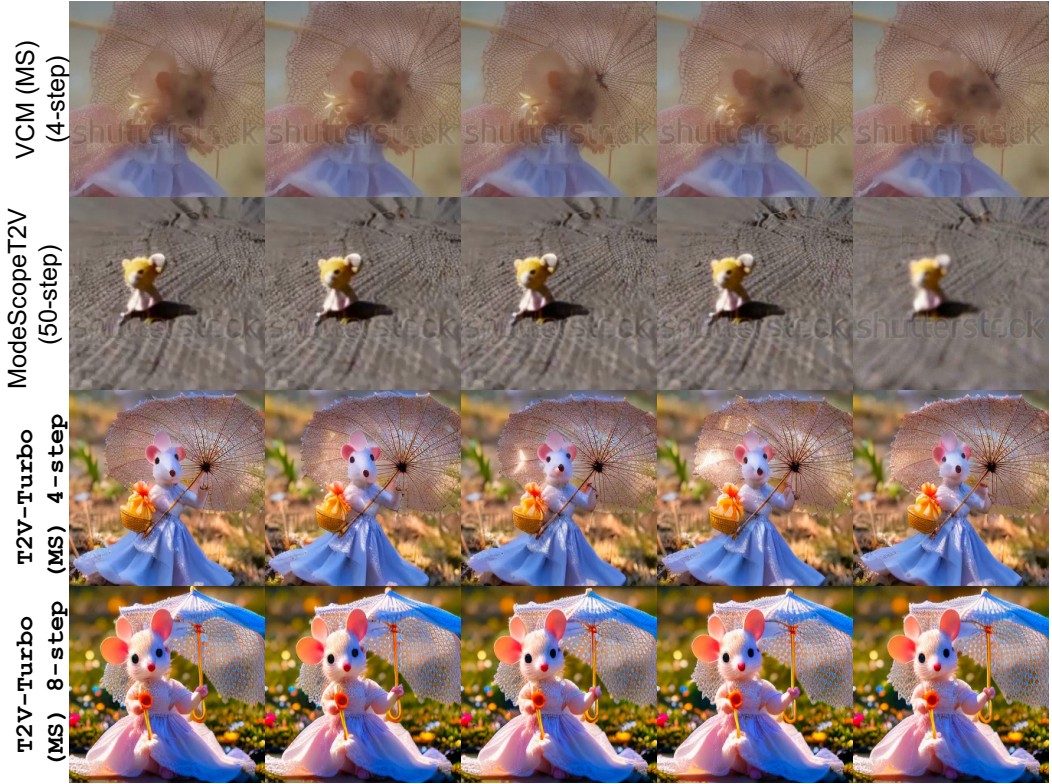

*pop art style, A photo of an astronaut riding a horse in the forest. There is a river in front of them with water lilies.*

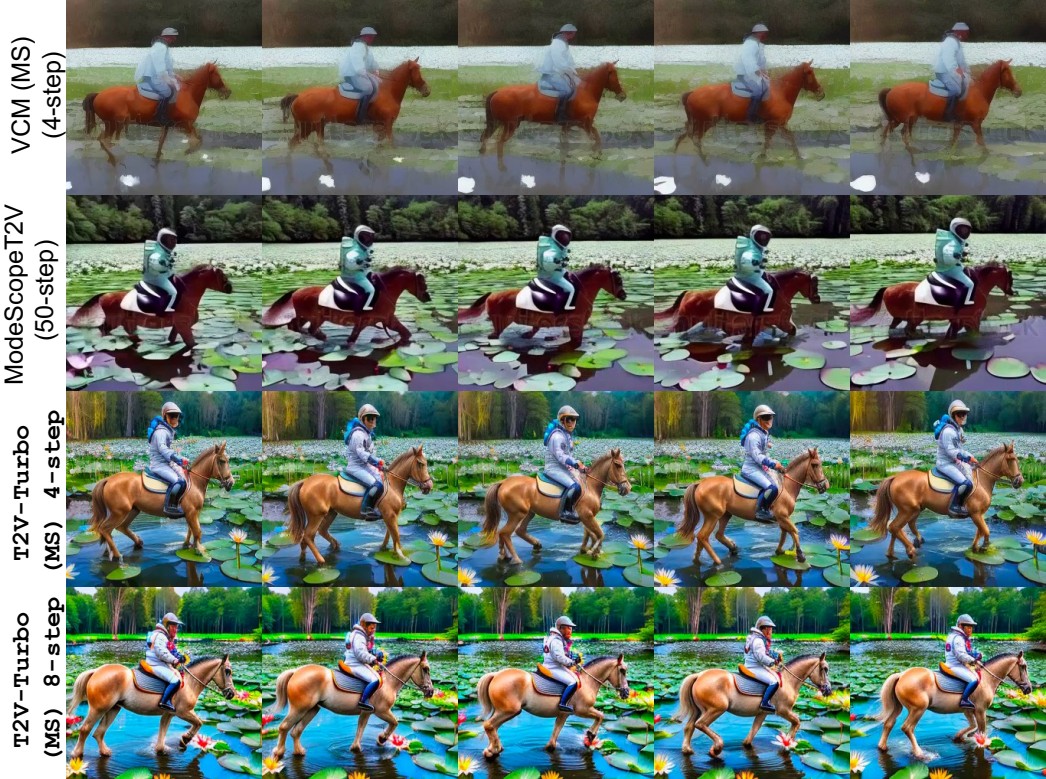

Figure 9: Additional qualitative comparison results for our T2V-Turbo (MS).

*Mickey Mouse is dancing on white background*

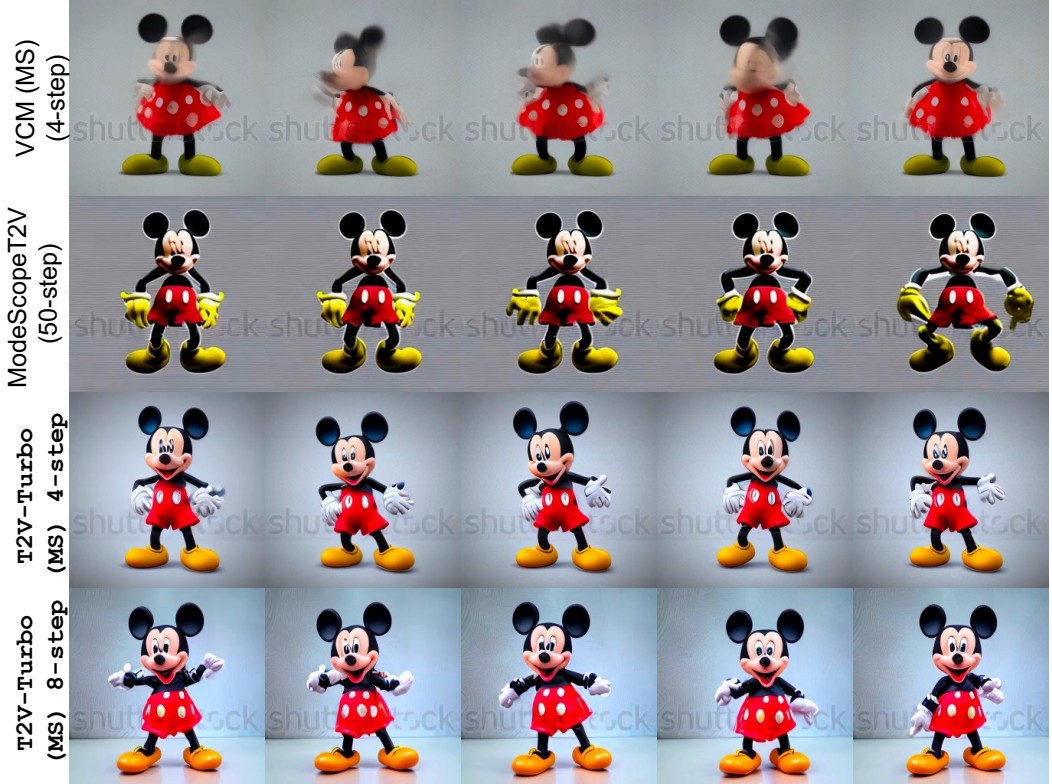

*a man looking at a distant mountain in Sci-fi style*

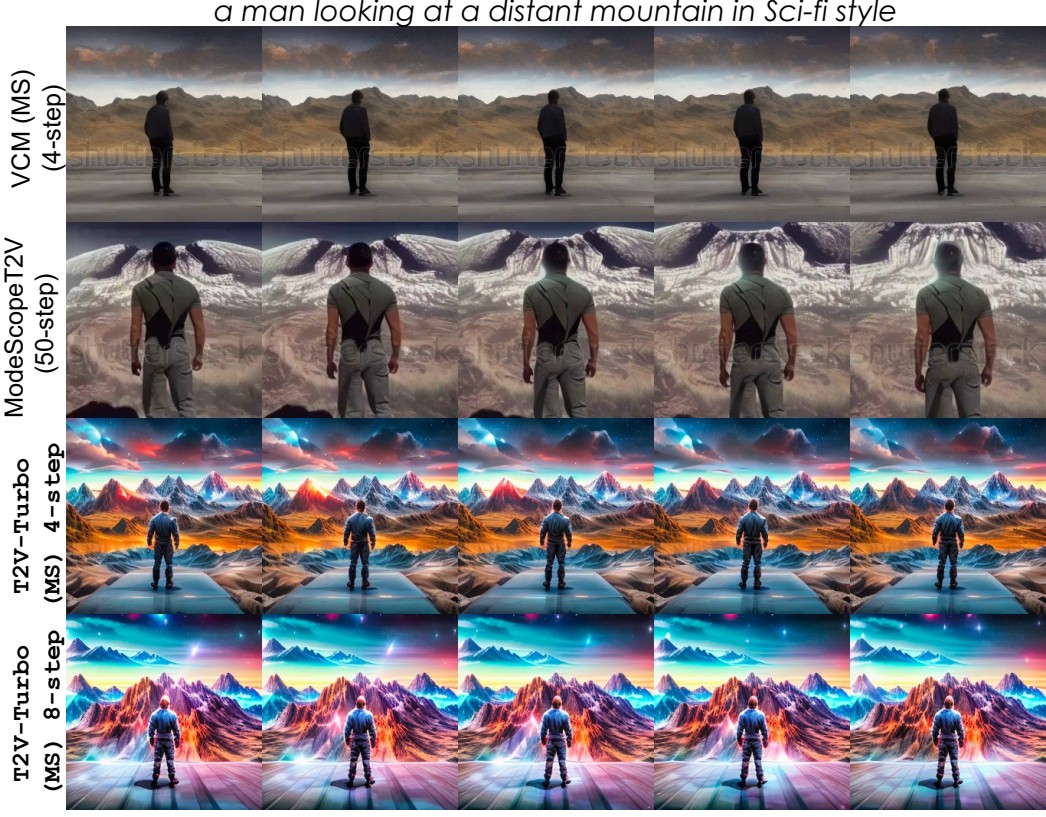

Figure 10: Additional qualitative comparison results for our T2V-Turbo (MS).

# H Comparing videos generated by `T2V-Turbo` and VCM + $\mathcal{R}_{\mathbf{img}}$

Please click to play videos in **Adobe Acrobat**.

**Prompt**: *A panda standing on a surfboard in the ocean in sunset.*

`T2V-Turbo` (VC2)                    VCM (VC2) + $\mathcal{R}_{\mathrm{img}}$

**Analysis** The panda on the **right** is instead **sitting** on the surfboard.

**Prompt**: *A raccoon is playing the electronic guitar.*

`T2V-Turbo` (VC2)                    VCM (VC2) + $\mathcal{R}_{\mathrm{img}}$

**Analysis** The **right** video fails on **playing the electronic guitar**.

**Prompt**: *A motorcycle accelerating to gain speed.*

`T2V-Turbo` (VC2)                    VCM (VC2) + $\mathcal{R}_{\mathrm{img}}$

**Analysis** The motorcycle on the **right** is actually moving backward.

**Prompt**: *A squirrel eating a burger.*

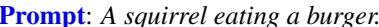

T2V-Turbo (VC2)                    VCM (VC2) + $\mathcal{R}_{\text{img}}$

**Analysis** The squirrel on the **right** looks more like it is **holding** a burger.

**Prompt**: *A Mars rover moving on Mars.*

T2V-Turbo (VC2)                    VCM (VC2) + $\mathcal{R}_{\text{img}}$

**Analysis** The hills on the **right** in the background also move.

**Prompt**: *A horse galloping across an open field.*

T2V-Turbo (VC2)                    VCM (VC2) + $\mathcal{R}_{\text{img}}$

**Analysis** Another horse suddenly runs into the scene of the **right** video.

**Prompt**: *A black vase.*

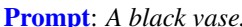

T2V-Turbo (VC2)                                    VCM (VC2) + $\mathcal{R}_{\text{img}}$

**Analysis** The **right** video shows **two vases** instead of one.

**Prompt**: *Happy dog wearing a yellow turtleneck, studio, portrait, dark background.*

T2V-Turbo (VC2)                                    VCM (VC2) + $\mathcal{R}_{\text{img}}$

**Analysis** The dog on the **right doesn't look happy**.

