# OpenReview forum: "T2V-Turbo: Breaking the Quality Bottleneck of Video Consistency Model with Mixed Reward Feedback"
_NeurIPS.cc/2024/Conference — NeurIPS 2024 poster_

### Official Review · Reviewer_Kq5y · 2024-07-02

**Soundness:** 3
**Presentation:** 3
**Contribution:** 3
**Rating:** 4
**Confidence:** 4

**Summary:**

The paper presents T2V-Turbo, a training strategy where additional reward models are introduced during the consistency distillation process to enhance the T2V consistency model's quality. With such enhancement, the trained model achieves favorable results in both VBench and human evaluations.

**Strengths:**

1. The paper is presented clearly and easy to follow;
2. The supplementary material provides abundant background information and results;
3. According to the figures, the proposed method significantly boosts the video quality.

**Weaknesses:**

1. Regarding the technical contribution, the proposed method seems like a straightforward extension of the video consistency model. The idea of adding direct supervision on the clean samples in consistency distillation has also been explored in previous works, e.g., [1];
2. It's hard to evaluate the motion quality improvements according to the static frames in the paper. It's highly recommended to include the corresponding mp4 files;
3. I wonder which reward model contributes more to quality improvement from Fig.4 row 1 and Fig.4 row 3? This cannot be inferred from Tab.2 and it's recommended to present some qualitative results in the ablation study;
4. The paper lacks in-depth discussions on the choice and influence of the image/video reward models. For instance, would it be better to CLIP/GAN-discriminator compared to the human preference score? Why do MS and VC2 perform differently when combined with InternVidS2?  Section 4.3 only provides empirical results.

[1] Adversarial Diffusion Distillation

**Questions:**

1. Why use different video reward models for ModelScope and VideoCrafter?
2. What leads to different hyperparameter choices in L164-165?

**Limitations:**

The authors have discussed the limitations.

---

> ### Author Rebuttal · Authors · 2024-08-06
>
> We thank the reviewer for the constructive feedback!
>
> > Regarding the technical contribution, the proposed method seems like a straightforward extension of the video consistency model. The idea of adding direct supervision on the clean samples in consistency distillation has also been explored in previous works, e.g., [1] Adversarial Diffusion Distillation (ADD).
>
> We would like to clarify that ADD is not based on consistency distillation. Instead, ADD's distillation loss corresponds to score distillation sampling [2]. Moreover, adversarial training is prone to instability [3] and might require massive hyperparameter tuning [4]. In contrast, our methods optimize towards differentiable RMs enjoying a stable training process. To the best of our knowledge, we are the first to add direct supervision on the clean samples when distilling from a video diffusion generator. Additionally, we are the first to learn video generators from the feedback of video-text models.
>
> [2] Poole et al. "Dreamfusion: Text-to-3d using 2d diffusion." ICLR 2023.
>
> [3] Yue et al. "On the algorithmic stability of adversarial training." NeurIPS 2021
>
> [4] Pang et al. "Bag of tricks for adversarial training." ICLR 2021
>
> > It's highly recommended to include the corresponding mp4 files
>
> We appreciate the suggestions! We promise to include all corresponding mp4 files in our revised manuscript and create a website to better present the videos generated by our methods.
>
> >  I wonder which reward model contributes more to quality improvement from Fig.4, row 1, and row 3?
>
> For Fig. 4, the image-text RM HPSv2.1 contributes more to our T2V-Turbo's (row 3) quality improvement over the baseline VCM (row 1). In the attached PDF, we empirically show that incorporating feedback from the $\mathcal{R}\_\text{vid}$ improves the video quality.
>
> > This cannot be inferred from Tab.2 and it's recommended to present some qualitative results in the ablation study
>
> First, we would like to clarify that Table 2 provides strong quantitative evidence for the effectiveness of each RM. For both VC2 and MS variants, leveraging feedback from $\mathcal{R}\_\text{img}$ alone (VCM + $\mathcal{R}\_\text{img}$) matches the **Quality Score** of our T2V-Turbo, indicating the similarly high visual quality of the generated videos. However, VCM + $\mathcal{R}\_\text{img}$ still falls behind our T2V-Turbo in terms of **Semantic Score**. Further incorporating feedback from $\mathcal{R}\_\text{vid}$ can bridge this gap, leading to better text-video alignment.
>
> The attached PDF corroborates the results in Table 2 with video examples. Specifically, it compares videos generated by VCM (VC2) + $\mathcal{R}\_\text{img}$ and our T2V-Turbo (VC2). The results show that while the visual quality of both methods is generally similar, additional feedback from $\mathcal{R}_\text{vid}$ significantly enhances the text-to-video alignment in T2V-Turbo.
>
> > The paper lacks in-depth discussions on the choice and influence of the image/video RMs. For instance, would it be better to CLIP/GAN-discriminator compared to the HPS?
>
> We thank the reviewer for the insightful question! Firstly, we would like to remind the reviewer that we have experimented with additional image-text RMs, including PickScore and ImageReward, as detailed in Appendix E. We qualitatively show that incorporating reward feedback from any of these image-text RMs leads to quality improvement over the baseline VCM (VC2). It is worth noting that HPSv2.1 and PickScore are fine-tuned from CLIP with human preference data. Therefore, learning from CLIP might not lead to better performance compared to learning from image-text RMs.
>
> We did not experiment with a GAN-discriminator, as training a GAN-discriminator can suffer from instability and may require substantial hyperparameter tuning. However, given the success of ADD, we acknowledge the potential benefit of learning from a GAN-discriminator. Consequently, we consider this a promising direction for future work.
>
> Lastly, we promise to include more in-depth discussions in our revised manuscript on the choice and influence of image/video RMs.
>
> > Why do MS and VC2 perform differently with InternVidS2?
>
> Firstly, we would like to emphasize that when training T2V-Turbo (MS) with InternVid2 S2, T2V-Turbo (MS) still shows improvement over both VCM (MS) and VCM (MS) + $\mathcal{R}\_\text{img}$. However, we acknowledge that ViCLIP is more effective in enhancing T2V-Turbo (MS) compared to InternVid2 S2, as demonstrated in Tables 2 and 3. In contrast, ViCLIP and InternVid2 S2 work similarly well for T2V-Turbo (VC2).
>
> We conjecture that this difference might be due to the quality disparity between the teacher models ModelScopeT2V and VideoCrafter2. For example, ModelScopeT2V suffers from low resolution, and the generated videos contain watermarks, which may limit VCM (MS)'s ability to learn from InternVid2 S2. Nonetheless, further tuning of the $\beta_\text{vid}$ parameter might help improve the performance of T2V-Turbo (MS) with InternVid2 S2.
>
> >  Why use different video RMs for ModelScope and VC2?
>
> Our main purpose is to **examine our methods with a diverse set** of $\mathcal{R}\_\text{vid}$. As we did not have access to a video-text RM trained to reflect human preference on video, we instead chose to experiment with video foundation models, including ViCLIP and InternVid2 S2. Note that we perform an ablation study and report results with both ViCLIP and InternVid2 S2 in Table 3.
>
> > What leads to different hyperparameter choices in L164-165?
>
> Compared to VideoCrafter2, ModelScopeT2V has not been trained on high-quality video or image datasets. Additionally, ModelScopeT2V suffers from low resolution, and the generated videos contain watermarks. Thus, ModelScope suffers from lower video quality. As a result, T2V-Turbo (MS) requires larger weighting parameters $\beta_\text{img}$ and $\beta_\text{vid}$ to get stronger external supervision from the RMs to improve its generation quality.

---

> ### Author Response · Authors · 2024-08-12
> **Follow-up the discussion**
>
> Dear Reviewer Kq5y,
>
> Your feedback has been invaluable in helping us clarify, improve, and refine our work. We have diligently addressed your comments in our response, made every effort to dispel misunderstandings, and provided various video examples for qualitative comparisons to demonstrate the effectiveness of our method.
>
> We kindly ask you to revisit our paper in light of our response and consider whether the changes and clarifications we have provided might warrant a reconsideration of your rating.
>
> Best regards,
>
> The Authors

---

> ### Author Response · Authors · 2024-08-13
> **New Results for Your Review!**
>
> Dear Reviewer Kq5y,
>
> We would like to kindly invite you to review the results we have submitted during the rebuttal period. We have thoroughly addressed your concerns and included videos corresponding to the static frames in our paper. Thanks to the support of our Area Chair, we are able to share an [anonymous website](https://spangled-blanket-128.notion.site/Qualitative-results-of-T2V-Turbo-1290f9ec0fb34685918438d7ba590e83#69781d1bb2a048c69ce6c51441ecde50) containing all these videos. We hope that you will reconsider your ratings in light of these new results : )
>
> Best regards,
>
> The Authors

---

> ### Comment · Reviewer_Kq5y · 2024-08-14
> **Re: Rebuttal**
>
> I appreciate the authors' detailed rebuttal and the additional results. They address some of my concerns.
>
> However, my major concern about the technical contribution still exists. Especially when comparing Fig. 2 in this paper and Fig. 2 in VideoLCM, the major technical contribution is adding a reward loss on the clean samples, which seems straightforward without too many challenges. Therefore, I tend to maintain my original score.

---

> > ### Author Response · Authors · 2024-08-14
> > **Clarification on the technical contribution**
> >
> > Dear Reviewer Kq5y,
> >
> > Thank you for your response. We want to further **clarify our technical contributions**.
> >
> > First, **we regret not emphasizing the technical challenge of learning from a video-text reward model (RM) sufficiently** in our original manuscript. Unlike learning from an image-text RM, obtaining feedback from a video-text RM $\mathcal{R}\_\text{vid}$ demands significantly more memory. For instance, using models like ViCLIP and InternVid2 S2 requires sampling a batch size of 8 frames from video clips. Since we work with a latent video generator, we must **enable gradients** during the decoding of these video frames from latent vectors to allow for passing from the $\mathcal{R}\_\text{vid}$ to the video generator. Consequently, this computational process becomes nearly impossible to fit within a 40GB A100 GPU if we also need to pass gradients through an iterative sampling process.
> >
> > To tackle this challenge, our method cleverly takes advantage of the single-step generation arising from consistency distillation, which is crucial for learning from $\mathcal{R}_\text{vid}$. Notably, even by operating with single-step generation, we almost fully utilize the 40GB memory of the A100 GPU.
> >
> > Additionally, we want to emphasize that **the technical simplicity of our method should not be seen as a drawback**. On the contrary, being technically simple yet highly effective is a distinct advantage of our approach.
> >
> > In this paper, we address the core challenges of video generation: 1) improving generation quality, 2) reducing inference time, and 3) alleviating the intensive memory and computational cost. Empirically, we achieve significant results by outperforming SOTA video-generation systems on standard benchmarks. **We hope the reviewer can evaluate our paper based on our core scientific contribution**.
> >
> > Looking forward to your response.
> >
> > Best regards,
> >
> > The Authors

---

### Official Review · Reviewer_61hZ · 2024-07-12

**Soundness:** 4
**Presentation:** 4
**Contribution:** 2
**Rating:** 6
**Confidence:** 3

**Summary:**

This paper presents a distillation method for text-to-video models. In short, it builds upon latent consistency models (more specifically, adapting the paper "Reward Guided Latent Consistency Distillation" from image to video models). The method involves the usual consistency model objective, in addition to that feedback from both an image-text reward model (e.g. HPSv2.1) and a video-text model (e.g. InternVideo) is used to enhance the quality of the generated videos. Overall, the proposed approach not only is a able to few-step video generation (4/8), but is also outperforming existing open-source video generators such as ModelScope and VideoCrafter.

**Strengths:**

The biggest strength of the paper would definitely be the strong results. Having a 4 step model that is able to achieve results comparable/better to 50 step models is indeed commendable.

In addition to this, the evaluation seems quite thorough: there are quantitative results on V-Bench with several categories, user studies comparing against standard video models, and also ablation studies which makes the paper quite sound.

Finally, the presentation of the paper is also quite clear, everything is easy to grasp and understandable.

**Weaknesses:**

The most obvious weakness about this paper is that it is a direct application of Li et al. ("Reward Guided Latent Consistency Distillation") to video models. As far as I can understand, the changes involve replacing SD2.1 with VideoCrafter/ModelScope and incorporating an additional video-text RM in addition to the image-text RM. Similarly, the main difference I can see with Wang et al. 2023 (VideoLCM) is the utilization of the reward objectives.

Given that previous works have shown the efficacy of Latent Consistency Models for video generation, and the benefits of reward models for LCM distillation in images, it's not entirely novel to combine these 2 ideas into a single distilled video model.

[Minor] I don't find too many qualitative examples, and I couldn't find more videos in the supplementary (apart from the code). It would definitely be nicer if there were more examples that were provided.

**Questions:**

Overall, I definitely lean towards accepting the paper. Despite being "obvious", I think that having an open-weight model+implementation for this would definitely help research in the field. However, I would really like the authors to point out if I have missed out details in their contributions or any other novel aspects of their work, since that is the major shortcoming I see currently.

**Limitations:**

Limitations are sufficiently discussed.

---

> ### Author Rebuttal · Authors · 2024-08-06
>
> We are grateful for the reviewer's positive feedback on our work. Please find our detailed response below.
>
> > I would really like the authors to point out if I have missed out details in their contributions or any other novel aspects of their work.
>
> We appreciate the opportunity to clarify our contributions. We would like to emphasize the importance of our mixture of RMs design, which enables us to achieve significant empirical results even without access to a $\mathcal{R}_\text{vid}$ trained to reflect human preference.
>
> Specifically, our method leverages feedback from both an image-text RM $\mathcal{R}\_\text{img}$ and a video foundation model $\mathcal{R}\_\text{vid}$, such as ViCLIP and InternVid2 S2. This combination allows our T2V-Turbo to break the quality bottlenecks in the video consistency model, resulting in both fast and high-quality video generation. Notably, the 4-step generations from our T2V-Turbo achieve the SOTA performance on VBench, surpassing proprietary models, including Gen-2 and Pika.
>
> Additionally, our ablation study in Table 2 empirically provides valuable scientific insights into the effectiveness of different RMs. While leveraging feedback from $\mathcal{R}\_\text{img}$ alone (VCM + $\mathcal{R}\_\text{img}$) is sufficient to match the visual quality (**Quality Score**) of our T2V-Turbo, the additional feedback from $\mathcal{R}\_\text{vid}$ further enhances text-video alignment, resulting in higher **Semantic Score**. Qualitative evidence is provided in the attached PDF. To the best of our knowledge, we are the first to improve video generation using the feedback from video-text RMs. We believe that future advancements in video-text RMs will further enhance the performance of our methods.
>
> Our joint training pipeline is computationally efficient. For example, InstructVideo requires over 40 hours of training to align a pretrained T2V model. In contrast, our method reduces training time to less than 10 hours, achieving a fourfold increase in efficiency in wall-clock training time.
>
> Lastly, our method is broadly applicable to a diverse set of prompts. We have empirically shown that our approach outperforms existing methods on comprehensive benchmarks, including VBench and EvalCrafter. In contrast, previous methods, such as InstructVideo, conduct experiments on a limited set of user prompts, most of which are related to animals.
>
> > I don't find too many qualitative examples, and I couldn't find more videos in the supplementary (apart from the code). It would definitely be nicer if there were more examples that were provided.
>
> Thank you for your suggestions! We have now included more videos in the attached PDF. We promise to include more videos in our revised manuscript and create a website to better present the videos generated by our methods.

---

> > ### Comment · Reviewer_61hZ · 2024-08-11
> > **Thanks for the clarification**
> >
> > I thank the authors for providing additional videos, and also clarifying the contributions of the method. Based on this, I am now raising my score.

---

> > > ### Author Response · Authors · 2024-08-12
> > > **Thank you!**
> > >
> > > Dear Reviewer 61hZ,
> > >
> > > Thank you for raising your score! Your feedback has been invaluable in helping us improve the presentation of our work. We are pleased that our rebuttal has clarified the contributions of our work.
> > >
> > > Thanks and best regards,
> > >
> > > The Authors

---

### Official Review · Reviewer_MMwk · 2024-07-13

**Soundness:** 2
**Presentation:** 3
**Contribution:** 2
**Rating:** 4
**Confidence:** 4

**Summary:**

This paper aims to achieve a video consistency model with both fast and high-quality generation. Specifically, the authors introduce T2V-Turbo, which integrates feedback from a mixture of differentiable reward models into the consistency distillation (CD) process of a pre-trained T2V model. The differentiable reward models consists of an image-text reward model and a video-text reward model. Experiment results verify the effectiveness of proposed method.

**Strengths:**

- The paper is easy to follow and understand.

- The idea is reasonable and simple.

- Experiment results are good.

**Weaknesses:**

- The novelty is limited. The paper seems to simple combine the video consistency model and the differentiable reward models. Different from previous works, the authors utilize a mixture of reward models, where a video-text reward model is additionally added to encourage the diffusion model to better model the temporal dynamics.  However, to me, this contribution is quite incremental.

- For the paper presentation, I think it is better to show what the text prompt is in Fig. 1 to help reader better understand the difference between different models.

- For the quantitative comparison, it can be seen that the proposed method actually doesn't perform the best for most of the evaluation metrics. I am not sure if comparing total score (i.e., a weighted sum of Quality Score and Semantic Score) only is enough to show the superiority of proposed method.

**Questions:**

See the weakness part.

**Limitations:**

The authors adequately addressed the limitations.

---

> ### Author Rebuttal · Authors · 2024-08-06
>
> We thank the reviewer for the constructive comments!
>
> >The paper seems to simple combine the video consistency model (VCM) and the differentiable reward models
>
> We emphasize that our method is NOT a simple combination of VCM and differentiable RM. Previous works focus on aligning a pretrained DM to the preference given by RMs. In contrast to previous methods, e.g., InstructVideo, aligning a pretrained T2V model by backpropagating gradients through the memory-intensive iterative sampling process, our method cleverly leverages the single-step generation that naturally arises from consistency distillation. By optimizing the rewards of the single-step generation, our method avoids the highly memory-intensive issues associated with passing gradients through an iterative sampling process.
>
> Additionally, our joint optimization technique is notably computationally efficient. Previous approaches, such as InstructVideo, require over 40 hours of training to align a pretrained T2V model. In contrast, our method reduces training time to less than 10 hours, achieving a fourfold increase in efficiency in wall-clock training time.
>
> Empirically, we have shown that our approach outperforms existing methods on comprehensive benchmarks, including VBench and EvalCrafter. In contrast, previous reward learning methods for video generation, such as InstructVideo, conducted experiments on a limited set of user prompts, most of which were related to animals.
>
> > the authors utilize a mixture of reward models, where a video-text reward model is additionally added to encourage the diffusion model to better model the temporal dynamics. However, to me, this contribution is quite incremental.
>
> We would like to clarify the contribution of our mixture of RMs design, which enables us to break the quality bottleneck in VCM without access to a $\mathcal{R}\_\text{vid}$ trained to mirror human preference. By learning from an image-text RM $\mathcal{R}\_\text{img}$ and a video foundation model $\mathcal{R}\_\text{vid}$, such as ViCLIP and InternVid2 S2, we achieve both fast and high-quality video generation. Notably, the 4-step generations from our T2V-Turbo achieve the SOTA performance on VBench, surpassing proprietary models, including Gen-2 and Pika.
>
> Additionally, our ablation study in Table 2 further provides valuable scientific insights into the effectiveness of different RMs. While leveraging feedback from $\mathcal{R}\_\text{img}$ alone (VCM + $\mathcal{R}\_\text{img}$) is sufficient to match the visual quality (**Quality Score**) of our T2V-Turbo, the additional feedback from $\mathcal{R}\_\text{vid}$ further enhances text-video alignment, resulting in higher **Semantic Score**. Qualitative evidence is provided in the attached PDF. To the best of our knowledge, we are the first to improve video generation using feedback from video-text RMs. We believe that future advancements in video-text RMs will further enhance the performance of our methods.
>
> > I think it is better to show what the text prompt is in Fig. 1 to help reader better understand the difference between different models.
>
> We thank the reviewer for the suggestion. We will include the corresponding prompts in our revised manuscript. The prompt for the top two and bottom two rows are
> 1. With the style of low-poly game art, A majestic, white horse gallops gracefully across a moonlit beach.
> 2. Kung Fu Panda posing in cyberpunk, neonpunk style
>
> >   For the quantitative comparison, it can be seen that the proposed method actually doesn't perform the best for most of the evaluation metrics. I am not sure if comparing total score (i.e., a weighted sum of Quality Score and Semantic Score) only is enough to show the superiority of proposed method.
>
> First, we emphasize that our automatic evaluation results in Table 1 are corroborated by the human evaluations in Fig 3, where the 4-step generation from both T2V-Turbo (VC2) and T2V-Turbo (MS) are preferred over the 50-step generations from their teacher VideoCrafter2 and ModelScopeT2V.
>
> Second, we highlight that the **Total Score, Quality Score, and Semantic Score are sufficient proxies for human preference**. Consider the comparison between VideoCrafter2 and our T2V-Turbo (VC2). Figure 3 indicates that human annotators favor the 4-step generation from our T2V-Turbo (VC2) in terms of **Visual Quality**, **Text-Video Alignment**, and **General Preference**. These preferences align with the higher **Quality Score**, **Semantic Score**, and **Total Score** of our T2V-Turbo (VC2), thereby validating these metrics' effectiveness in reflecting our method's superiority.
>
> Lastly, we address why our T2V-Turbo does not perform the best across all evaluation metrics yet still achieves the highest overall scores. We extracted the performance of VideoCrafter2 and our T2V-Turbo (VC2) from Table 1 of our paper. Although VideoCrafter2 scores slightly higher on 5 out of 7 dimensions constituting the **Quality Score** and 4 out of 9 dimensions constituting the **Semantic Score**, the differences are not significant. In contrast, our T2V-Turbo (VC2) significantly outperforms VideoCrafter2 in metrics such as `Dynamic Degree`, `Image Quality`, and `Multiple Objects`, contributing to its overall superiority.
>
> | Models|Total Score|Quality Score|Subject Consist.|BG Consist.|Temporal Flicker|Motion Smooth.|Aesthetic Quality|Dynamic Degree|Image Quality|
> |-|-|-|-|-|-|-|-|-|-|
> | VideoCrafter2| 80.44|82.20|**96.85**|**98.22**|**98.41**|**97.73**|**63.13**|42.50|67.22|
> | $\texttt{T2V-Turbo}$ (VC2)|**81.01**|**82.57**|96.28|97.02|97.48|97.34|63.04|**49.17**|**72.49**|
>
> | Models|Semantic Score|Object Class|Multiple Objects|Human Action|Color|Spatial Relation.|Scene|Appear. Style|Temporal Style|Overall Consist.|
> |-|-|-|-|-|-|-|-|-|-|-|
> |VideoCrafter2|73.42|92.55|40.66|95.00|**92.92**|35.86|55.29|**25.13**|**25.84**|**28.23**|
> |$\texttt{T2V-Turbo}$ (VC2)|**74.76**|**93.96**|**54.65**|**95.20**|89.90| **38.67**|**55.58**|24.42|25.51| 28.16|

---

> > ### Author Response · Authors · 2024-08-12
> > **Follow-up the discussion**
> >
> > Dear Reviewer MMwk,
> >
> > We greatly appreciate your insightful feedback, which has significantly contributed to the clarity and enhancement of our work. We have carefully addressed your comments in our response, worked to resolve any misunderstandings, and included multiple video examples for qualitative comparisons to illustrate the effectiveness of our method.
> >
> > We kindly request that you revisit our paper in light of our response and clarifications, and consider whether these updates might lead to a reevaluation of your rating.
> >
> > Best regards,
> >
> > The Authors

---

> > ### Comment · Reviewer_MMwk · 2024-08-13
> > **Re: Rebuttal**
> >
> > Thanks for the authors' detailed response. However, the response still does not clarify my concerns about the novelty. To me, the method proposed in this paper is a combination of VCM and differentiable RM. I understand that compared with differentiable RM methods, it is different as it introduces VCM instead of using conventional diffusion models. I also understand that combining the two techniques does bring benefits and achieve superior results. However, I didn't see this paper solving any novel technical issues when combining the two techniques. To me, the engineering significance of this paper is greater than its technological innovation. Thus, I tend to keep my original rating.

---

> > > ### Author Response · Authors · 2024-08-13
> > > **Clarification on our technical significance**
> > >
> > > Dear Reviewer MMwk,
> > >
> > > Thank you for your response. We want to clarify the **technical issues** we solved in this paper and emphasize our technical significance.
> > >
> > > 1. Traditional reward learning methods for video generation, such as InstructVideo, suffer from intensive memory costs due to backpropagating gradients through a diffusion model (DM)'s iterative sampling process. Our method solves this issue by leveraging the single-step generation arising from consistency distillation.
> > >
> > > 2. Our method significantly reduces the training time required to align a video generation model, achieving a 4x increase in efficiency in wall-clock training time compared to InstructVideo.
> > >
> > > We also wish to address a **major understanding**. Our approach **does NOT simply replace the DM with a VCM** from a conventional reward learning method. A straightforward combination of VCM and differentiable RM requires either 1) distilling a VCM from a pretrained DM and subsequently aligning the VCM with a differentiable RM, or 2) aligning a DM with a differentiable RM before distilling a VCM from the aligned DM.
> > >
> > > Neither of these sequential methods achieves the same memory reduction as our approach because they still involve backpropagating gradients through the iterative sampling processes of either the DM or the VCM. Furthermore, these methods are not as computationally efficient as ours due to their two-phase training pipelines.
> > >
> > > We sincerely appreciate your feedback and hope that this clarification will encourage you to reconsider your evaluation of our work.
> > >
> > > Best regards,
> > >
> > > The Authors

---

> > > > ### Comment · Reviewer_MMwk · 2024-08-13
> > > > **Re: technical significance**
> > > >
> > > > Thanks for the authors' further clarification.
> > > >
> > > > After reading the authors' clarification, I further checked the paper. My concern still exists. To me, the joint optimization adopted in the paper is also straightforward. I didn't see any gap or technical difficulty to perform the joint optimization. As claimed in Fig. 2 and Eq. (10) of the paper, the paper just simply integrates reward feedback into the VCD procedures (i.e., adding the RM loss terms to the original VCD loss term) .
> > > >
> > > > Thus, I still think the paper or the method lacks technical novelty. I will keep my original rating.

---

> ### Author Response · Authors · 2024-08-13
> **Further clarification**
>
> Dear Reviewer MMwk,
>
> Thanks for your response.
>
> Firstly, we would like to emphasize that the technical simplicity of our method should not be seen as a drawback. On the contrary, **being technically simple yet highly effective is a distinct advantage of our approach**. In this paper, we address the core challenges of video generation: 1) improving generation quality, 2) reducing inference time, and 3) alleviating the intensive memory and computational cost when aligning a video generator.
>
> Current state-of-the-art proprietary video generation systems, such as Gen-2 and Pika, require **several minutes** to produce a short video clip. In contrast, our T2V-Turbo can generate high-quality videos **within 5 seconds**, making it significantly more suitable for real-time applications. T2V-Turbo achieves both fast and high-quality video generation. As validated on VBench, its 4-step generation process surpasses the performance of proprietary systems like Gen-2 and Pika, which rely on extensive resources. To the best of our knowledge, we are the first to simultaneously address these two contradictory aspects—**speed and quality**—within the same video-generation framework.
>
> Secondly, we would like to reiterate the significance of our mixture of RMs design, which allows us to break the quality bottleneck in VCM without access to a $\mathcal{R}\_\text{vid}$ trained to mirror human preference. Our ablation study in Table 2 empirically provides valuable scientific insights into the effectiveness of different RMs.
>
> While leveraging feedback from $\mathcal{R}\_\text{img}$ alone (VCM + $\mathcal{R}\_\text{img}$) is sufficient to match the visual quality (**Quality Score**) of our T2V-Turbo, the additional feedback from $\mathcal{R}\_\text{vid}$ further enhances text-video alignment, resulting in higher **Semantic Score**. We provide qualitative evidence in the attached PDF. To the best of our knowledge, we are the first to improve video generation with feedback from video-text RMs. We believe that future advancements in video-text RMs will further improve the performance of our methods.
>
> Thank you again for your time and effort in providing valuable feedback on our work. We hope our clarifications will lead to a more favorable evaluation of our contribution.
>
> Best,
>
> The Authors

---

> > ### Author Response · Authors · 2024-08-14
> > **On the technical difficulty of learning from a video-text RM**
> >
> > Dear Reviewer MMwk,
> >
> > We would like to bring your attention to **the technical difficulty of learning from a video-text RM**. We regret not emphasizing this challenge sufficiently in our original manuscript. Unlike learning from an image-text RM, obtaining feedback from a video-text RM $\mathcal{R}\_\text{vid}$ demands significantly more memory. For instance, using models like ViCLIP and InternVid2 S2 requires sampling a batch size of 8 frames from the generated video clips. Since we work with a latent video generator, we must enable gradients while decoding these video frames from latent vectors to allow for passing from  $\mathcal{R}\_\text{vid}$ to the video generator. Consequently, this computational process becomes nearly impossible to fit within a 40GB A100 GPU if we also need to pass gradients through an iterative sampling process.
> >
> > Our method addresses this challenge by cleverly leveraging single-step generation from consistency distillation. This approach is crucial, as even with single-step generation, we nearly max out the 40GB memory of the A100 GPU—let alone handling gradients through an iterative sampling process.
> >
> > As the discussion period draws to a close, we kindly ask the reviewer to **reconsider their rating of our paper by focusing on our core scientific contributions**.
> >
> > Best regards,
> >
> > The Authors

---

### Official Review · Reviewer_x94N · 2024-07-14

**Soundness:** 3
**Presentation:** 4
**Contribution:** 4
**Rating:** 7
**Confidence:** 4

**Summary:**

The paper introduces T2V-Turbo to enhance the quality of video consistency models in text-to-video generation. The authors address the slow sampling speed of diffusion-based T2V models and the low quality of generated video by integrating feedback from a mixture of differentiable reward models into the consistency distillation (CD) process of a pre-trained T2V model. This integration allows for the optimization of single-step generations, bypassing the memory constraints of backpropagating gradients through iterative sampling processes. T2V-Turbo demonstrates significant improvements in both speed and quality, achieving high performance on the VBench benchmark and surpassing leading models such as Gen-2 and Pika.

**Strengths:**

- **Efficiency and Quality**: T2V-Turbo achieves impressive results with 4-step generations, offering a tenfold acceleration in inference speed while improving video quality compared to 50-step DDIM samples from teacher models.
 - **Comprehensive Evaluation**: The authors conduct extensive experiments, including automatic evaluations on VBench and human evaluations with 700 prompts from EvalCrafter, to validate the effectiveness of T2V-Turbo.

**Weaknesses:**

- **Technical contribution**: The proposed approach is simply a mixture of previous works, i.e., consistency distillation and reward feedback learning, which deflates the technical contribution of the paper. However, I acknowledge that this is one of the pioneering approaches in text-to-video generation.
- **Reward models**: While the proposed method heavily depends on the reward models, the reward models that the paper used are actually not designed to function as reward models. This is because of the absence of video-text reward models in comparison to image generation (as the author mentioned in their limitations). However, it is a more right approach to first design a good reward model for video generation than developing a video generation model to align with such reward models.

**Questions:**

- Regarding reward models for video generation. Could the author provide more details about how the reward models should be designed in order to obtain better video generation or evaluating video generation models?
- It seems like only consistency distillation for VC2 degrades the performance (Table 1), while the score remains the same for ModelScope. What if we do not use consistency distillation, and only use reward feedback learning for VC2 or MS? Then this may have a higher score than T2V-Turbo? Suppose it is true, then what if we distill after the feedback learning? To summarize my points, does the joint training of feedback learning and consistency distillation is better than sequential fine-tuning?

**Limitations:**

The paper highlights the limitation of relying on existing video-text reward models that are not explicitly trained to reflect human preferences on video-text pairs. Instead, the authors use video foundation models like ViCLIP and InternVid S2 as substitutes. While incorporating feedback from these models has enhanced T2V-Turbo’s performance, the authors acknowledge that the development of more advanced video-text reward models could further improve the results. Additionally, the complexity of integrating mixed reward feedback into the CD process could hinder the broader adoption and scalability of the approach.

---

> ### Author Rebuttal · Authors · 2024-08-06
>
> We thank the reviewer for the positive feedback on our work! Please find our detailed response below.
>
> > Q1: The proposed approach is simply a mixture of previous works, i.e., consistency distillation and reward feedback learning
>
> We emphasize that our method is NOT simply combining the video consistency model and reward feedback learning. Previous methods, such as InstructVideo [1], require backpropagating gradients through an iterative sampling process, which can lead to substantial memory costs. In contrast, our method cleverly leverages the single-step generation arising from consistency distillation. By optimizing the rewards of the single-step generation, our method avoids the highly memory-intensive issues associated with passing gradients through an iterative sampling process.
>
> Additionally, we would like to emphasize the contribution of our mixture of RMs design, which enables us to break the quality bottleneck in VCM without access to a $\mathcal{R}\_\text{vid}$ trained to mirror human preference. By learning from an image-text RM $\mathcal{R}\_\text{img}$ and a video foundation model $\mathcal{R}\_\text{vid}$, such as ViCLIP and InternVid2 S2, we achieve both fast and high-quality video generation. Notably, the 4-step generations from our T2V-Turbo achieve the SOTA performance on VBench, surpassing proprietary models, including Gen-2 and Pika. To the best of our knowledge, we are the first to improve video generation using feedback from video-text RMs. We believe that future advancements in video-text RMs will further enhance the performance of our methods.
>
> [1] Yuan et al., InstructVideo: Instructing Video Diffusion Models with Human Feedback. CVPR 2024
>
> > Q2: However, it is a more right approach to first design a good reward model for video generation than developing a video generation model to align with such reward models.
>
> We fully recognize the importance of designing a good video-text RM $\mathcal{R}\_\text{vid}$. However, creating an effective $\mathcal{R}\_\text{vid}$ might require long-term efforts. A video is more complex than an image due to its additional temporal dimension. We envision that it requires multiple iterations to derive an effective $\mathcal{R}\_\text{vid}$. Specifically, one iteration involves 1) Training $\mathcal{R}\_\text{vid}$ by collecting new preference data from a video generator and 2) Training the video generator to align with $\mathcal{R}\_\text{vid}$. **Our method provides an efficient way to accomplish the second step.**
>
> On the other hand, even without a $\mathcal{R}_\text{vid}$ trained to reflect human preferences on video-text pairs,  we highlight that our method can still enhance the generation quality of a T2V model by aligning it with video-text foundation models, such as ViCLIP and InternVid2 S2.
>
> > Q3: Could the author provide more details about how the reward models should be designed in order to obtain better video generation or evaluating video generation models?
>
> We thank the reviewer for the question. In addition to our response to Q2,  we conjecture a $\mathcal{R}_\text{vid}$ effective for training video generators might need to be finetuned from existing video foundation models, such as ViCLIP and InternVid2 S2. This approach mirrors the success seen with image-text RMs, including HPSv2, ImageReward, PickScore, and AestheticScore, which have proven effective for training image generators. These models are finetuned from image-text foundation models, such as CLIP and BLIP, using human preference data.
>
> Moreover, learning a multi-dimensional reward to reflect fine-grained human preferences might also be helpful. For example, each dimension of the reward vector could be trained to reflect visual quality, transition dynamics, text-to-video alignment, etc.
>
> > Q4: Does the joint training of feedback learning and consistency distillation is better than sequential fine-tuning?
>
> First, sequential fine-tuning is computationally more expensive than joint training. According to the InstructVideo paper, their reward feedback learning phase costs more than 40 hours. If we add the distillation time cost further, the total time cost for sequential fine-tuning can easily exceed 50 hours. Conversely, our joint training only requires less than 10 hours, representing a fivefold reduction in terms of training wall-clock time.
>
> Second, as mentioned in Sec. 1 of our paper, finetuning a diffusion T2V model towards a differentiable RM requires backpropagating gradients through the diffusion model's iterative sampling process. Therefore, calculating the full reward gradient is prohibitively expensive, resulting in substantial memory costs. Conversely, our method leverages the single-step generation that arises naturally from computing the CD loss, effectively bypassing the memory constraints.
>
> > Q5: It seems like only consistency distillation for VC2 degrades the performance (Table 1)
>
> In terms of the performance degradation of distilling VC2, we conjecture it can be alleviated by performing full model training, or further optimizing the hyperparameters related to learning the LoRA weights.

---

> > ### Comment · Reviewer_x94N · 2024-08-11
> >
> > Thank you for the detailed rebuttal and the extensive qualitative examples provided. I acknowledge the computational burden involved in T2V generation and the meticulous nature of evaluating video generation models. This paper contributes valuable strategies for improving T2V generation and offers a robust framework for evaluating video generation models, which will be beneficial for other researchers in the field. After careful consideration, I will maintain my original score.

---

> > > ### Author Response · Authors · 2024-08-11
> > >
> > > Dear Reviewer x94N,
> > >
> > > Thank you again for your positive feedback on our work!
> > >
> > > Best regards,
> > >
> > > The Authors

---

### Official Review · Reviewer_niV6 · 2024-08-07

**Soundness:** 4
**Presentation:** 3
**Contribution:** 3
**Rating:** 7
**Confidence:** 4

**Summary:**

The paper proposes T2V-Turbo, a model aiming to achieve both fast and high-quality text-to-video generation by breaking the quality bottleneck of a video consistency model (VCM). It integrates mixed reward feedback from one image and one video reward model into the consistency distillation process of a teacher T2V model. The 4-step generations from T2V-Turbo outperform SOTA methods on the VBench benchmark and are favored by humans over the 50-step DDIM samples from the teacher model, achieving over ten-fold inference acceleration with quality improvement.

**Strengths:**

- The paper is well-structured and clearly presents the problem, the proposed method, the experimental setup, and the results. The use of figures and tables helps to illustrate the concepts and results.
- The automatic evaluation results on the VBench benchmark and human evaluation results with the 700 prompts from EvalCrafter demonstrate the effectiveness of the proposed method, with T2V-Turbo outperforming baseline methods and proprietary systems in terms of total score and human preference.
- The ability to generate high-quality videos quickly has significant impacts in various fields, such as digital art and visual content creation, and sets a new benchmark for future research in T2V synthesis.

**Weaknesses:**

- The method mainly combines the consistency distillation in the Video Consistency Model (VCM) with multiple reward models. Although it has achieved good results in solving the existing problems of the T2V model, this combination is relatively conventional and may be somewhat lacking in originality.
- The citation format of this paper should adhere to the NeurIPS standard, the authors misuse \citet throughout the paper, especially in Sections 2 and 5, making it hard to read smoothly.

**Questions:**

- In Line 228, the authors write “InternVid2 S2 outperforms ViCLIP in several zero-shot video-text retrieval task”.  While in Table 1, we know that ModelScopeT2V (MS) has less total score than VideoCrafter2 (VC2), which means that MS is a relatively weaker model than VC2, then why choose ViCLIP as the video RM for MS rather than using a stronger InternVid2 S2?

**Limitations:**

The authors have adequately addressed the limitations of the work in the paper. They acknowledge the lack of an open-sourced video-text reward model trained to reflect human preferences on video-text pairs and discuss the potential use of a more advanced video reward model in the future. They also mention the concerns about misinformation and deepfakes raised by the ability to create highly realistic synthetic videos and commit to installing safeguards when releasing the models, such as requiring users to adhere to usage guidelines.

---

> ### Author Response · Authors · 2024-08-07
> **Rebuttal by Authors**
>
> We thank the reviewer for the positive feedback on our work! Please find our detailed feedback below.
>
> > The method mainly combines the consistency distillation in the Video Consistency Model (VCM) with multiple reward models. Although it has achieved good results in solving the existing problems of the T2V model, this combination is relatively conventional and may be somewhat lacking in originality.
>
> We would like to highlight that our method cleverly leverages the single-step generation arising from consistency distillation. By optimizing the rewards of the single-step generation, our method avoids the highly memory-intensive issues associated with passing gradients through an iterative sampling process. In contrast, previous methods, such as InstructVideo [1], require backpropagating gradients through diffusion model's iterative sampling process, resulting in substantial memory costs.
>
> Additionally, we emphasize the importance of our mixture of RMs design, which enables us to break the quality bottleneck in VCM without access to a $\mathcal{R}\_\text{vid}$ trained to mirror human preference. By learning from an image-text RM $\mathcal{R}\_\text{img}$ and a video foundation model $\mathcal{R}\_\text{vid}$, such as ViCLIP and InternVid2 S2, we achieve both fast and high-quality video generation. Notably, the 4-step generations from our T2V-Turbo achieve the SOTA performance on VBench, surpassing proprietary models, including Gen-2 and Pika. To the best of our knowledge, we are the first to improve video generation using feedback from video-text RMs. We believe that future advancements in video-text RMs will further enhance the performance of our methods.
>
> > The citation format of this paper should adhere to the NeurIPS standard, the authors misuse \citet throughout the paper, especially in Sections 2 and 5, making it hard to read smoothly.
>
> We thank the reviewer for pointing out the issues! We will make sure to fix the citation issues in our revised manuscript!
>
> > why choose ViCLIP as the video RM for MS rather than using a stronger InternVid2 S2?
>
> Our main purpose is to **examine our methods with a diverse set** of $\mathcal{R}\_\text{vid}$. As we did not have access to a $\mathcal{R}\_\text{vid}$ trained to reflect human preference on video, we instead chose to experiment with video foundation models, including ViCLIP and InternVid2 S2. Notably, we conduct a comprehensive ablation study in Table 3, presenting results for our T2V-Turbo (VC2) and T2V-Turbo (MS) when setting $\mathcal{R}\_\text{vid}$ to both ViCLIP and InternVid2 S2. This approach allows for a thorough assessment of our methods across different $\mathcal{R}\_\text{vid}$.

---

### Author Rebuttal · Authors · 2024-08-06

We appreciate the reviewers for their time and constructive feedback on our work. We have responded to individual reviews below and would like to highlight additional qualitative results in the attached PDF. **Please download and open it with Adobe Acrobat** to click and play the videos.

The attached PDF **corroborates the results in Table 2** with video examples. To demonstrate the effectiveness of our mixture of RMs, we compare 8 pairs of videos generated by our $\texttt{T2V-Turbo}$ and $\text{VCM}$ + $\mathcal{R}\_\text{img}$. Due to space constraints, we focus on the VC2 variants and did not include results for $\text{VCM}$ and $\text{VCM}$ + $\mathcal{R}_\text{vid}$, as our $\texttt{T2V-Turbo}$'s results are significantly better. Additionally, we include videos corresponding to Figure 4 of our paper to better showcase the superiority of our $\texttt{T2V-Turbo}$ over its teacher model and the baseline $\text{VCM}$.

We would also like to re-iterate the contributions of our methods:
1. To the best of our knowledge, we are the first to improve video generation using feedback from video-text RMs.
2. We emphasize the importance of our mixture of RMs design, which enables us to break the quality bottlenecks in VCM without access to a $\mathcal{R}\_\text{vid}$ trained to mirror human preference. The 4-step generations from our $\texttt{T2V-Turbo}$ set the SOTA performance on VBench, surpassing proprietary models like Gen-2 and Pika.
3. Our training pipeline is NOT a simple combination of VCM and differentiable reward. Our method cleverly leverages the single-step generation arising from consistency distillation, avoiding the need to backpropagate gradients through the memory-intensive iterative sampling process required by traditional methods, such as InstructVideo.
4. Our method is notably **computationally efficient**. For example, InstructVideo requires over 40 hours of training to align a pretrained T2V model. In contrast, our method reduces training time to less than 10 hours, achieving a fourfold increase in efficiency in wall-clock training time.
5. Our method is **broadly applicable to a diverse set of prompts**. We have empirically shown that our approach outperforms existing methods on comprehensive benchmarks, including VBench and EvalCrafter. In contrast, InstructVideo's experiments are conducted on a limited set of user prompts, most of which are related to animals.

---

### Comment · Area_Chair_TJhJ · 2024-08-07
**Anonymous website link with generated videos**

Dear reviewers,

As requested in the reviews, the authors, after consultation with me, have shared an anonymous website containing video results. The videos are available at https://spangled-blanket-128.notion.site/Qualitative-results-of-T2V-Turbo-1290f9ec0fb34685918438d7ba590e83#69781d1bb2a048c69ce6c51441ecde50. Please consider this part of the rebuttal response.

Best,
Your AC

---

### Decision · Program_Chairs · 2024-09-25

**Decision:**

Accept (poster)

**Comment:**

The paper received two borderline rejects, one weak accept and two accepts. The concerns were primarily centered around lack of technical novelty since the paper's summary can be distilled to "applying reward models to distillation". But all the reviewers point out the impressive results with the approach using very few steps. The paper applies multiple image and video reward models during consistency distillation to avoid memory bottlenecks. It also showcases that a true video reward model (which don't exist publicly) is not needed, and using video-text models as reward models also boosts performance. The AC agrees with the majority of reviews and feels that given the simplicity of implementation and the impressive results, the paper merits acceptance.